
# Constraints on global aerosol number concentration, SO₂ and condensation sink in UKESM1 using ATom measurements

Ananth Ranjithkumar[1], Hamish Gordon[2,1], Christina Williamson[3,4], Andrew Rollins[3], Kirsty J. Pringle[1], Agnieszka Kupc[4,5], Nathan Luke Abraham[6,7], Charles A. Brock[4], Kenneth S. Carslaw[1]

[1]School of Earth and Environment, University of Leeds, LS2 9JT, United Kingdom
[2]Engineering Research Accelerator and Centre for Atmospheric Particle Studies, Carnegie Mellon University, Pittsburgh, PA, 15213, USA
[3] Cooperative Institute for Research in Environmental Sciences, University of Colorado, Boulder, CO 80309, USA
[4]NOAA Chemical Sciences Laboratory, Boulder, CO 80305, USA
[5]Faculty of Physics, Aerosol Physics and Environmental Physics, University of Vienna, 1090 Vienna, Austria
[6]National Centre for Atmospheric Science, UK
[7]Department of Chemistry, University of Cambridge, Cambridge, UK

*Correspondence to*: Ananth Ranjithkumar (eeara@leeds.ac.uk)

**Abstract**

Understanding the vertical distribution of aerosol helps to reduce the uncertainty in the aerosol lifecycle and therefore in the estimation of the direct and indirect aerosol forcing. To improve our understanding, we use measurements from four deployments of the Atmospheric Tomography (ATom) field campaign (ATom1-4) which systematically sampled data over the Pacific and Atlantic Oceans with near pole-to-pole coverage. We evaluate the UK Earth system model (UKESM1) against ATom observations in terms of joint biases in the vertical profile of three variables related to new particle formation: total particle number concentration ($N_{Total}$), sulphur dioxide ($SO_2$) mixing ratio and the condensation sink. The $N_{Total}$, $SO_2$ and condensation sink are interdependent quantities and have a controlling influence on the vertical profile of each other. Improving only one of these quantities in comparison with observations can lead to a misleading impression that overall model performance has improved. Analysing $N_{Total}$, $SO_2$ and condensation sink simultaneously helps reduce the probability of getting the right answer for the wrong reasons. The model's condensation sink is within a factor of 2 of observations, but the $N_{Total}$ and $SO_2$ shows larger biases mainly in the tropics and high latitudes. Algorithmic improvements to the model and perturbations to key atmospheric processes help reduce tropospheric model biases consistently. We performed a series of model sensitivity tests to identify atmospheric processes that have the strongest influence on overall model performance ($N_{Total}$, $SO_2$ and condensation sink simultaneously). In the boundary layer (which we define in this study as below 1 km altitude) and lower troposphere (1-4 km) inclusion of a boundary layer nucleation scheme





(Metzger et al., 2010), which is switched off in the default version of UKESM, is critical to obtaining better agreement with observations. However, in the mid (4-8 km) and upper troposphere (>8 km), sub-3 nm particle

growth, pH of cloud droplets, DMS emissions, upper tropospheric nucleation rate, $SO_2$ gas scavenging rate and cloud erosion rate are found to play a more dominant role. Analysing the data with altitude, we find that perturbations to boundary layer nucleation, sub 3 nm growth, cloud droplet pH and DMS emissions reduces the boundary layer and upper tropospheric model bias. We performed a combined simulation with all 4 perturbations included and found that the model's $N_{Total}$, $SO_2$ and condensation sink biases were reduced in most cases (up to a

50% reduction) in both the boundary layer and upper troposphere simultaneously. These perturbations are well-motivated in that they improve the physical basis of the model and are suitable for implementation in future versions of UKESM.

## 1 Introduction

Aerosols affect the global energy balance by directly scattering and absorbing solar radiation, and indirectly by their ability to act as cloud condensation nuclei (CCN), which changes the microphysical properties of clouds (Albrecht, 1989; Twomey, 1977). The direct and indirect effect aerosols have on climate has been identified as the largest source of uncertainty in the assessment of anthropogenic forcing (Bellouin et al., 2020; Carslaw et al., 2013; Myhre et al., 2013). The direct radiative forcing by aerosol particles is dependent on the scattering and absorption

of solar radiation, which in turn is dependent on aerosol optical properties like their size, shape and refractive index. The indirect radiative forcing is dependent on aerosol particles forming or behaving as CCN (or ice nuclei), which is controlled by the hygroscopicity and aerosol size distribution at cloud base (1– 3 km). There are still gaps in our knowledge of atmospheric processes that control the spatial, temporal and size distribution of aerosols in the atmosphere. Atmospheric aerosol concentrations depend on their sources; primary (emissions) and secondary (new

particle formation and particle growth), their sinks (scavenging, wet and dry deposition) and transport through the atmosphere (Merikanto et al., 2009). Thus, the different atmospheric processes that have a controlling influence on the aerosol distribution throughout the atmosphere must be better understood.

Global-scale measurements of aerosol microphysical properties are needed to evaluate general circulation models (GCMs). Satellite measurements have extensive global coverage, but they cannot detect particles smaller than about

100 nm diameter. In-situ aircraft measurements give more detailed information about the full size distribution, chemical composition and radiative properties of aerosol particles. In past studies (Dunne et al., 2016; Ekman et





al., 2012; Watson-Parris et al., 2019) global models have been compared against measurement campaigns such as CARIBIC (Civil Aircraft for Regular Investigation of the Atmosphere Based on an Instrument) (Heintzenberg et al., 2011), ACE1 (First Aerosol characterization experiment) (Clarke et al., 1998), PEM Tropics (Pacific

Exploratory missions - Tropics) (Clarke et al., 1999), ARCTAS (Arctic Research of the composition of the troposphere from aircraft and satellites) (Jacob et al., 2010), PASE (Pacific Atmosphere Sulphur experiment) (Faloona et al., 2009), INTEX-A (Intercontinental chemistry transport experiment – North America) (Singh et al., 2006) and VOCALS (VAMOS Ocean-Cloud-Atmosphere-Land Study) (Wood et al., 2011). Each of these campaigns had goals to help us understand particle size distribution in the upper troposphere, the particle production

rate in cloud outflow regions, Arctic atmospheric composition, sulphur processing, tropospheric composition over land and clouds/precipitation in the south-eastern Pacific respectively. The measurements from these campaigns were used to identify atmospheric processes that help constrain the particle size distribution in global climate models like MIT-CAM3 (Ekman et al., 2012) and ECHAM-HAM (Watson-Parris et al., 2019) with observations.

In this work, we compare in-situ aircraft observations conducted as part of the NASA Atmospheric Tomography Mission (ATom) (Wofsy et al., 2018) to a global climate model (UKESM1) to better quantify the model biases in particle number concentration, $SO_2$ and the condensation sink. The ATom campaigns provide a representative continuous data set of daytime aerosol, gas and radical concentrations and properties by continuously sampling the atmosphere vertically and spatially over a vast region of the marine free troposphere. This single global dataset was

obtained between 2016 and 2018 during four campaigns sampling each of the four seasons. During these campaigns, a large aerosol and gas instrument payload was deployed on the NASA DC-8 aircraft for systematic sampling of the atmosphere spanning altitudes between 0.2 km and 12 km, and spatially it encompasses Pacific and Atlantic oceans with near pole-to-pole coverage. This data has been used recently (Williamson et al., 2019) to highlight the importance of new particle formation to CCN concentration in the upper and free troposphere, and

highlights severe deficiencies in the ability of state of the art global chemistry climate models to capture new particle formation, particle growth and aerosol vertical transport accurately.

The ATom data have also been used in previous work to address biases in the vertical profile of sea salt and black carbon in the Community Earth System Model (CESM) and to better understand the in-cloud removal of aerosols

by deep convection (Yu et al., 2019). Other studies used the measurements to address uncertainties associated with the life cycle of organic aerosol in the remote troposphere (Hodzic et al., 2020) and to investigate the mechanisms of new particle formation in the tropical upper troposphere (Kupc et al., 2020). The measurements have also shed



light on the global distribution of biomass burning aerosol (Schill et al., 2020), brown carbon (Zeng et al., 2020) and DMS oxidation chemistry (Veres et al., 2020).


Although the ATom dataset is extensive and provides important information about aerosol number and gas concentrations (Williamson et al., 2019; Wofsy et al 2018), there are some challenges when comparing it to a GCM. A single data point sampled represents a point in the atmosphere defined by the latitude, longitude, altitude and time the data was collected. The UKESM output is, however, an average over a broad horizontal grid box of ~135km across, and it is usually temporally averaged over a month. In previous studies (Schutgens et al., 2016) it has been shown that sampling errors can be minimized by averaging the observations over time and model errors can be reduced by using 4D model fields with high temporal resolution. In the first part of this paper, we evaluate UKESM at three-hour time resolution against observations and highlight some of the biases that exist in the model in different regions of Earth.


In the second part of this paper, we focus on trying to understand and reduce these biases. We focus on processes related to new particle formation, as this is the dominant source of aerosol number concentration globally (Gordon et al., 2017; Yu and Luo, 2009). Some model developments and a series of sensitivity simulations are performed to determine the source of the model-measurement bias. As well as resolving a bug in the model, we also address some of the deficiencies in the nucleation mode microphysics and the dependence of coagulation sink on particle diameter. The sensitivity tests comprise model simulations in which we perturb various parameters that control different atmospheric processes, one at a time.

In order to obtain physically motivated reductions in model bias, we evaluate the model simultaneously against three observed quantities related to new particle formation: total particle number concentration ($N_{Total}$), $SO_2$ mixing ratio and condensation sink (Adams and Seinfeld, 2002). The condensation sink is a measure of how rapidly condensable vapor molecules (in UKESM, sulphuric acid and secondary organic aerosol material) and newly formed molecular clusters are removed by the existing aerosol surface area. It is a loss term for new particles, while $SO_2$ is effectively a production term because it controls sulphuric acid vapour concentrations. Assessing the influence of model processes on only one of these quantities in one-at-a-time sensitivity tests can result in misleading or incomplete conclusions about model performance, because different atmospheric processes affect $N_{Total}$, $SO_2$ and the condensation sink to varying degrees and can be independent of each other. As an example, an atmospheric process like in-cloud production of sulphate aerosol can increase the condensation sink, which will





decrease the gas concentration of precursors such as sulphuric acid, $H_2SO_4$, for new particle formation, and then in

turn decrease $N_{Total}$. Perturbing atmospheric processes can also have a direct effect on the $SO_2$ mixing ratio which in turn can have an effect on the $H_2SO_4$ concentration. Improving the model-observation match to only one of $N_{Total}$, $SO_2$ and the condensation sink can result in a poorer match for the other two quantities. Therefore, it is important to identify atmospheric processes that reduce $N_{Total}$, $SO_2$ and condensation sink biases simultaneously.

**2. The ATom Dataset**

The main goal of the ATom campaign was to improve our scientific understanding of the chemistry and climate processes in the remote atmosphere over marine regions. In relation to aerosols, the campaign helps to quantify the abundance, distribution, composition and optical properties of aerosol particles in the remote atmosphere. This can help determine the source of these particles and evaluate the mechanism for formation and growth of new particles

to form CCN. The whole campaign used the NASA DC-8 research aircraft and was subdivided into four series of flights, ATom1 (August – September 2016), ATom2 (January – February 2017), ATom3 (September – October 2017) and ATom4 (April – May 2018). The flight path for each of the ATom deployments is shown in Figure 1. Measurements were made between ~0.18 km and ~12 km altitude, from the Antarctic to the Arctic, over the Atlantic and Pacific oceans. All of the data are publicly available (Wofsy et al., 2018).

We used the $SO_2$ data from ATom4 (the $SO_2$ data from ATom1-3 were not sensitive at concentrations less than 100ppt) and the particle number concentration data from ATom1, ATom2, ATom3 and ATom4. The instruments used to measure the aerosol size distribution from 2.7 nm to 4.8 μm are a nucleation-mode aerosol size spectrometer (NMASS) (Williamson et al., 2018), an ultra-high-sensitivity aerosol size spectrometer (UHSAS) and a laser aerosol spectrometer (LAS). The NMASS consists of five continuous laminar flow condensation particle counters

(CPCs) in parallel, with each CPC operated at different settings so as to detect different size classes (Brock et al., 2019; Williamson et al., 2018). During ATom 1, the cut-off sizes (probability of the particles at cut-off size to be detected is greater than 50%) for each of the CPCs were 3.2 nm, 8.3 nm,14 nm, 27 nm and 59 nm. From ATom 2 to ATom 4, additional cut-off sizes of 5.2, 6.9, 11, 20 and 38 nm were present. This setup helps establish the aerosol size distribution for particles smaller than 59 nm. The UHSAS measures particle number concentrations for

particles with diameter between 63 nm and 1000 nm (Kupc et al., 2018). The LAS efficiently measures particles between 120 nm and 4.8 μm. The POPS instrument was operated as a backup to detect coarse-mode particles (Gao et al., 2016).



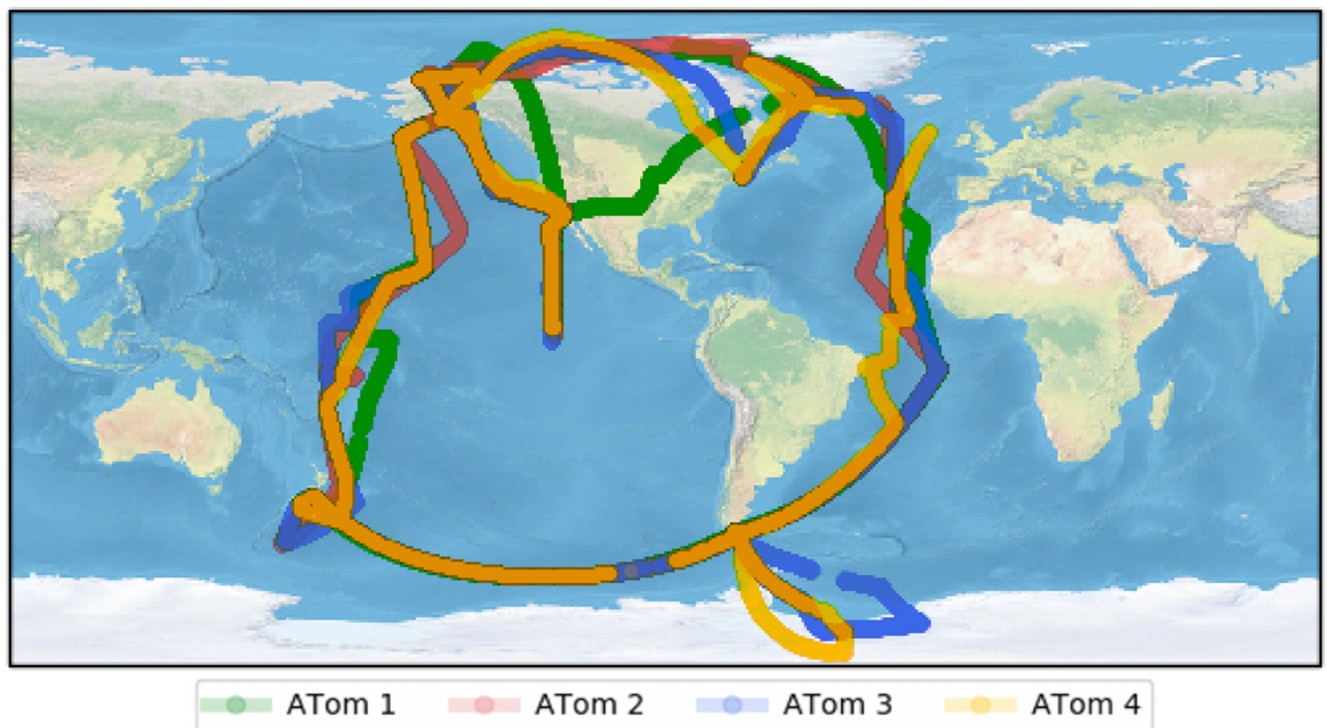

Figure 1: Flight tracks for NASA DC-8 for the 4 ATom campaigns: ATom1 (August – September 2016, green),
ATom2 (January – February 2017, red), ATom3 (September – October 2017, blue) and ATom4 (April – May 2018,
yellow)

The $SO_2$ measurements were obtained using the laser-induced fluorescence instrument (Rollins et al., 2016). $SO_2$
mixing ratios at high altitudes are quite low (between 1-10 parts per trillion). It is difficult to measure $SO_2$ mixing
ratio at low pressure with high precision. This instrument is capable of retrieving precise measurements of $SO_2$
concentration at pressures as low as 35 hPa making this instrument operable up to altitudes of 20km. The instrument
has a detection limit of 2 ppt (at a 10s measurement interval), and an overall uncertainty of $\pm(16\%+0.9\text{ppt})$.

## 3. Model Description

The model used in this work is the United Kingdom Earth system Model version 1 (UKESM1) (Mulcahy et al.,
2020; Sellar et al., 2019) in its atmosphere-only configuration (with fixed sea surface temperatures and prescribed



biogenic emissions from a fully coupled model simulation). The latest HadGEM3 global coupled (GC) climate configuration of the UK Met office was used to develop UKESM. HadGEM3 consists of the core physical dynamical processes of the atmosphere, land, ocean and sea ice systems (Ridley et al., 2018; Storkey et al., 2018; Walters et al., 2017). The UK's contribution to the Coupled Model Intercomparison Project Phase 6 (CMIP 6) (Eyring et al., 2015) is comprised of model simulations from the HadGEM3 and UKESM1 models.

Atmospheric composition is simulated with the chemistry-aerosol component of UKESM which is the UK Chemistry and Aerosol model (UKCA) (Morgenstern et al., 2009; O'Connor et al., 2014; Archibald et al., 2020). The aerosol scheme within UKCA is referred to as the Global Model of Aerosol Processes, GLOMAP-mode, (Mann et al., 2010; Mulcahy et al., 2020). It uses a two-moment pseudo-modal approach and simulates multicomponent global aerosol which includes sulphate, black carbon, organic matter and sea spray. Dust is simulated separately using a difference scheme (Woodward, 2001). GLOMAP-mode includes aerosol microphysical processes of new particle formation, condensation, coagulation, wet scavenging, dry deposition and cloud processing. The aerosol particle size distribution is represented using 5 log-normal modes: nucleation soluble, Aitken soluble, accumulation soluble, coarse soluble and Aitken insoluble, with their size ranges shown in Table A1 (Appendix A). UKCA is coupled to other modules in UKESM to handle tracer transport by convection, advection and boundary layer mixing. Originally in GLOMAP-mode, sulphate and secondary organic formation was driven by prescribed oxidant fields (Mann et al., 2010). However, in this study the UKCA chemistry and aerosol modules are fully coupled (Mulcahy et al., 2020).

The model can be run in different configurations (Walters et al., 2017), in this work we use the N96L85 configuration which is 1.875º x 1.25º longitude-latitude, corresponding to a horizontal resolution of approximately 135km. The model has 85 vertical levels up to an altitude of 85 km from the Earth's surface, with 50 levels between 0 and 18km, and 35 levels between 18 and 85 km. To compare the model against observations, we run the model in a nudged configuration, in which horizontal winds and potential temperature of the model are relaxed towards fields from the ERA–interim reanalysis fields (Dee et al., 2011; Telford et al., 2008). This helps to reproduce the same meteorological conditions at the exact time and location the measurements were performed, and to reduce model biases compared to free-running configurations (Kipling et al., 2013; Zhang et al., 2014). A relaxation time constant of 6 hours is chosen (equal to the temporal resolution of the reanalysis fields), and the nudging is applied between model levels 12 and 80. When comparing the model data to observations, the output fields from the model are retrieved at high temporal resolution (3-hourly output). This is done to reduce model sampling errors (Schutgens



et al., 2016). The diagnostics fields that we use for our analysis are total particle number concentration ($N_{Total}$),
      sulphur dioxide ($SO_2$) mixing ratio and condensation sink. These 4D diagnostics fields occupy significant disk
      space, and due to storage space constraints, we developed an online interpolator to process the model fields as and
      when they are output to give the value of the required diagnostics at the exact time and location where the
      measurement was obtained. To reduce sampling errors, 5-minute averages of the measurements were used in this
study. The interpolated diagnostic fields occupy less storage space and are retained for our analysis while the
      original large model field file is erased.

## 4. Evaluation of the baseline model

Figure 2 shows the simulated longitudinal mean fields of total particle number concentration ($N_{Total}$), $SO_2$ mixing
ratio and condensation sink from the atmosphere-only configuration of UKESM. The particle number
concentrations are much lower at the surface than the free and upper troposphere, mainly due to the stronger
      production rate of new particles via binary homogenous nucleation at higher altitudes. The highest zonal mean
      $N_{Total}$ concentration ($8 \times 10^4$ particles/cm$^3$ at STP) occurs at an altitude range of 12 to 16 km. At an altitude of 15
      km, most of the particles are present in the intertropical latitude band ($25^0N - 25^0S$). The $SO_2$ mixing ratio is
      maximum (>1000ppt) at the surface in the northern hemisphere because there are significant $SO_2$ sources from land
as a consequence of industrial activity. In the southern hemisphere, the $SO_2$ source is mainly from the oxidation of
      dimethyl sulphide emitted from the ocean. The $SO_2$ mixing ratio at high altitudes is substantial, with a simulated
      mixing ratio of ~50 pptv (at 15 km) in the tropics. A secondary peak in the mixing ratio of $SO_2$ occurs at 30 km
      altitude from the oxidation of carbonyl sulphide (we include the stratosphere up to 30km altitude in Figure 2 for
      completeness and the troposphere is the main focus of this study). The condensation sink is directly related to the
number of large particles present in the atmosphere, which provides a surface for the condensation of condensable
      vapours like $H_2SO_4$. Large particles are typically present at a lower altitude; this leads to a higher condensation
      sink close to the surface, where its maximum value (when longitudinally averaged) is ~0.01 s$^{-1}$ (i.e., lifetime of
      condensable vapours before condensation is ~100 s). The minimum in the condensation sink is around $5 \times 10^{-5}$ s$^{-1}$,
      in the upper troposphere. A low condensation sink at a higher altitude increases the lifetime and mixing ratio of
condensable vapours like $H_2SO_4$ which is an important factor in the rapid formation of new particles at these
      altitudes.







Figure 2: Global longitudinal mean vertical profile of the simulated a) total particle number concentration ($N_{Total}$),
b) $SO_2$ mixing ratio and c) condensation sink from the default version of our model. In this figure, we show
altitudes up to 30km, and our model top is 85km, but our analysis focuses on the troposphere. The black dashed
line represents the tropopause height.

To compare the model with ATom data, we use high temporal resolution 4D model output data along the flight
track. Vertical profiles of the model data (along the flight track) and observations are shown in Figure 3,4 and 5.
The default version of the model shows substantial biases when compared to observations. On investigating these
biases, we discovered a bug in the subroutine in which the tendency in $H_2SO_4$ concentration in the chemistry scheme





was calculated. The chemistry and aerosol processes in the model are handled using the operator splitting technique, where the usual timestep for chemical reactions is 1 hour and the algorithm that handles the chemistry introduces sub-steps where necessary. Microphysical processes (nucleation, condensation and coagulation) are treated on a

separate 4-minute-long sub-timestep within the 1-hour chemistry timestep. The $H_2SO_4$ concentration is updated on every microphysics time step, and this was incorrectly implemented: the production of sulphuric acid from $SO_3$ on the microphysics time step was missing and the sulphuric acid was being produced only at the beginning of every chemistry time step. This resulted in an excess $H_2SO_4$ concentration at the beginning of every chemistry time step, but no production of $H_2SO_4$ later in the timestep. Nucleation is a very non-linear process, and so the high initial

$H_2SO_4$ concentration resulted in an excessive number of small particles being produced via nucleation. We resolved this bug and used this corrected version, which we refer to as the 'baseline' version, as the starting point for our sensitivity analysis in Section 6. The released version of UKESM, which we started with, does not contain the bug-fix and was used in CMIP6 experiments (Eyring et al., 2015). In this study we refer to this version of the model as the 'default' version.

Figure 3 compares the simulated and measured vertical profile of $N_{Total}$ and the model-measurement normalised mean bias factor (NMBF) (defined in equation 1) (Yu et al., 2006) for the default and baseline simulations. The global data is divided into three regions: the tropics (25N-25S), mid-latitudes (25N – 60N, 25S - 60S) and high latitudes (60N - 90N, 60S – 90S). The default version of UKESM is shown in red, the baseline version of UKESM in green and the ATom measurements in black. The magnitude of the model bias is quantified by the value

1+|NMBF|, which is the factor by which the model over- or underestimates the observations.

$$NMBF = \begin{cases} \frac{\sum M_i}{\sum O_i} - 1 = \frac{\bar{M}}{\bar{O}} - 1, & if \bar{M} \geq \bar{O} \\ 1 - \frac{\sum O_i}{\sum M_i} = 1 - \frac{\bar{O}}{\bar{M}}, & if \bar{M} < \bar{O} \end{cases} \quad \text{....... (1)}$$

where $M$ indicates Model and $O$ is the observation. A positive NMBF indicates that the model prediction is higher than the measurements and a negative value indicates that the model is lower than the measurements.

The default model substantially overpredicts $N_{Total}$ (Figure 3) in the upper troposphere (>8 km), with a factor of 10-

15 overestimate at an altitude of 12 km in the tropics. In the lower free troposphere (between 1 km and 3 km) and boundary layer (<1 km), the model agrees well (NMBF ~ 0) with observations in the tropics. However, the model underestimates the observations by a factor of 3 in the mid and high latitudes. The baseline (bug-fixed) version of





the model shows biases a factor 5-10 lower in the upper troposphere than the default version, for the reasons explained above.

Figure 4 shows the vertical profile of $SO_2$ mixing ratio in the model. The baseline and default model versions have similar mixing ratios and are positively biased by approximately a factor 2-6 in the boundary layer regions of the tropics and midlatitudes. In the tropical upper troposphere, the model overpredicts $SO_2$ by up to a factor 2-6, while the biases in the upper tropospheric mid and high latitudes are negligible. It is expected that the baseline and default models perform similarly, as the bug-fix does not affect $SO_2$ mixing ratios.

Figure 5 shows the vertical profile of the condensation sink in the atmosphere. The condensation sink simulated by the default and baseline versions of the model both show positive and negative biases within a factor of 2 of the observations. Larger particles in the atmosphere contribute to the condensation sink and a higher concentration of these large particles would result in more available surface area for condensable vapours to condense. The bias when comparing the model to observations can be explained by uncertainties in primary aerosol/gas emissions or

other atmospheric processes. From the vertical profile it appears that the model either transports larger aerosol particles to the free troposphere or removes too little in precipitation.

We also note that we have lumped northern and southern hemispheric data for the mid and high latitudes. The magnitudes of $N_{Total}$, $SO_2$ and condensation sink are different in both hemispheres and we illustrate that in Appendix Figure A1. The vertical profiles of all three variables show similar biases in both the northern and southern

midlatitudes. In the high latitudes we see more substantial interhemispheric differences. The most notable are, a) $N_{Total}$ shows a factor of 5 underprediction in the northern high latitude boundary layer, with southern high latitude boundary layer showing good agreement with observations, b) The model predicts less than 1pptv $SO_2$ mixing ratio in the southern high latitudes with observation showing a mixing ratio of ~10ppt. We explore ways to reduce these biases in section 6 and 7.

From Figure 3, 4 and 5, an immediate result of the baseline model evaluation is that the too-high particle number concentration in the free and upper troposphere at tropical and mid-latitudes is qualitatively consistent with too-high $SO_2$ mixing ratios, but inconsistent with the too-high condensation sink. The possible reasons for the biases in $N_{Total}$, $SO_2$ and condensation sink is explored later in section 5.


## Vertical profile of Total particle number concentration

a)

| Tropics | Midlatitudes | Highlatitudes |

b)

| Tropics | Midlatitudes | Highlatitudes |

observation    baseline    default





Figure 3: a) The vertical profile of the total particle number concentration (at standard temperature and pressure (STP)) as observed (ATom1-4) and in the simulated data from the default and baseline (bug-fixed) configurations of the UKESM, b) The vertical profile of the normalised mean bias factor (NMBF) for the two configurations of the model. The vertical profiles have been provided for the tropics (25ºN-25ºS), Midlatitudes (25ºN-60⁰N and 25ºS-60ºS) and High latitudes (60ºN-90ºN and 60ºS-90ºS). In both a) and b) the bold line represents the median and the shaded region represents the corresponding interquartile range (25th and 75th percentile) in a 1km altitude bin.





# Vertical profile of SO₂ mixing ratio

a)

| Tropics | Midlatitudes | Highlatitudes |

b)

| Tropics | Midlatitudes | Highlatitudes |





Figure 4: a) The vertical profile of the $SO_2$ mixing ratio as observed (ATom4 (April – May 2018)) and in the simulated data from the default and baseline configurations of the UKESM, b) The vertical profile of the Normalised Mean Bias Factor (NMBF) for the two configurations of the model. The vertical profiles have been provided for the tropics (25°N-25°S), midlatitude (25°N-60°N and 25°S-60°S) and high latitudes (60°N-90°N and 60°S-90°S). In both a) and b) the bold line represents the median and the shaded region represents the corresponding interquartile range ($25^{th}$ and $75^{th}$ percentile) in a 1km altitude bin.





Figure 5: a) The vertical profile of the dry condensation sink in the atmosphere, as observed, and in simulated data from the default and baseline configurations of UKESM, b) The vertical profile of the Normalised Mean Bias Factor (NMBF) for the two configurations of the model. The vertical profiles have been provided for the tropics (25ºN-25ºS), Midlatitudes (25ºN-60ºN and 25ºS-60ºS) and High latitudes (60ºN-90ºN and 60ºS-90ºS). In both a)



and b) the bold line represents the median and the shaded region represents the corresponding interquartile range (25[th] and 75[th] percentile) in a 1km altitude bin.

## 5. Model sensitivity simulations and improvements to model microphysics

To investigate the potential causes of the model biases, we have identified several atmospheric processes that are expected to influence the vertical profile of the $N_{Total}$, $SO_2$ and condensation sink. The model simulations that we performed include a combination of direct perturbations to atmospheric processes and changes in model microphysics. The perturbations were applied globally, and we analyse model performance at different regions in the troposphere. A more complete method of sensitivity analysis is to consider the joint effect of a combination of

parameters on model performance, which has been done in the past with perturbed parameter ensemble studies (Lee et al., 2013; Regayre et al., 2018). The one-at-a time sensitivity tests that we carry out here help to determine which processes have the largest effect on model biases and this information can be used in ensemble studies in the future. The atmospheric processes which we have selected for this study along with the motivation for why we picked them is described from Section 5.1 to 5.5 and also summarised in Table 1. A more detailed analysis of the

effect of these model simulations on model biases is described in Section 6.

| Atmospheric process/parameter | Perturbation to parameter in UKESM |
|---|---|
| pH of cloud droplets | pH = 6 & 7 (default pH = 5) |
| Boundary layer nucleation  (Metzger et al., 2010) | BL_nuc & BL_nuc/10 |
| Condensation sink | condsink*5 & condsink*10 |
| Primary marine organic emissions | primmoc & primmoc*5 |
| Coagulation sink dependence on particle diameter | sub_3nm_growth represented using (Lehtinen et al., 2007) |
| DMS emissions | Seadms=1.0 (default = 1.7) |
| Binary $H_2SO_4$-$H_2O$ nucleation rate | Jveh/10 & Jveh/100 |
| $SO_2$ wet scavenging rate | csca*10 & csca*20 |
| Cloud erosion rate | dbsdtbs = 0 & $10^{-3}$ |
| Aerosol wet scavenging efficiency | rscav_ait = 0.3 & 0.7, rscav_accu = 0.7, rscav_coarse = 0.9 |
| Coagulation kernel | coag*5 |



Table 1: Overview of the atmospheric processes that we have chosen for one-at-a-time sensitivity tests and the magnitude of the perturbation/scaling applied.

## 5.1 Nucleation rate and nucleation-mode microphysics

*Binary homogeneous nucleation.* UKESM uses a binary neutral homogeneous $H_2SO_4$-$H_2O$ nucleation scheme
(Vehkamäki et al., 2002) throughout the atmosphere. The upper tropospheric positive biases in $N_{Total}$ which we see
from Figure 3 could be because of a high nucleation rate.  Therefore, we perform simulations where we reduce the
nucleation rate by a factor of 10 and 100 to assess its influence on the large bias in upper-tropospheric particle
number concentration. These perturbations to the nucleation rate could indirectly compensate for the biases in the
production rate of $H_2SO_4$ from $SO_2$ (which can affect the concentration of sulphuric acid in the atmosphere, which
affects new particle formation). It should be noted that the $H_2SO_4$-$H_2O$ nucleation scheme (Vehkamäki et al., 2002)
is an old scheme and the parameterised nucleation rates are valid only for a limited temperature range (230 K – 305
K). A new nucleation scheme (Määttänen et al., 2018) for the $H_2SO_4$-$H_2O$ system extended the validity range to
lower temperatures and a wider range of environmental conditions. Global particle number concentration for both
schemes were compared in that study (Määttänen et al., 2018) and the vertical profile of particle number
concentration was found to be slightly higher (by ~100 particles/cm$^3$) at lower altitude (between 300 hPa and 800
hPa), with particle number concentrations in the upper troposphere (>300 hPa) being almost identical. This
addresses the uncertainty associated with the Vehkamaki nucleation scheme for the $H_2SO_4$-$H_2O$ system at low
temperatures in the upper troposphere. However, this perturbation is not well-motivated by available nucleation
parameterizations but is intended only as a candidate for crude tuning to compensate for model biases.

*Boundary layer nucleation.* We incorporated a boundary layer nucleation (BLN) scheme (Metzger et al., 2010) to
account for a source of new particles in the boundary layer to address the model's  boundary layer negative bias
(Figure 5). Most of our measurements are over remote ocean and the scheme we use is dependent on oxidation
products from organics which, in our model, originate only from terrestrial vegetation. However, these organic
vapours or the nucleated particles are transported to the remote ocean and thereby affect the vertical profile. The
condensation sink is also affected by BLN since the new particles that are formed can grow to larger particles by
condensation of sulphuric acid and volatile organic compounds onto their surface (Pierce et al., 2012). We perform
one model simulation with boundary layer nucleation included and then one where the boundary layer nucleation
rate is reduced by a factor of 10. All of the oxidation products of volatile organic compounds (VOCs) are treated





similarly in the model and have been lumped into a tracer called 'Sec_org'. This could lead to biases in the BLN
rate and condensational particle growth rate since in reality the oxidation products of VOCs have different
volatilities which can nucleate and condense at different rates. Reducing this nucleation rate by a factor of 10
(Regayre et al., 2018; Yoshioka et al., 2019) was found to match better with observations.

*New particle growth.* We improved the handling of the growth of newly formed clusters in the model because the
initial stage of particle growth up to about 3 nm diameter is crucial to global CCN concentrations (Gordon et al.,
2017; Tröstl et al., 2016) and can affect the vertical profile of particle number concentration. Measurement of
particle growth rate at diameters smaller than 3 nm is difficult for most atmospheric instrumentation. This growth
of small particles is determined by competing processes where particles grow by condensation of vapour onto the
particle surface and are lost by coagulation with larger pre-existing particles (Pierce and Adams, 2007). Particle
growth is simulated explicitly for particle sizes larger than 3nm. However, for the sub-3nm size range, the growth
is represented implicitly by defining an effective rate of production of particles at 3 nm (accounting for competing
growth and loss processes). This rate is calculated using a parameterization (Kerminen and Kulmala, 2002):

$$J_{3nm} = J_{d_c} exp \left( \frac{CS(d_c)}{GR} \cdot d_c{}^2 \cdot \left( \frac{1}{3} - \frac{1}{d_c} \right) \right) \qquad …………….. (2)$$

where $J_{3nm}$ and $J_{dc}$ refer to the particle production rate at 3 nm and the critical size ($d_c$) respectively, CS ($d_c$) is the
coagulation sink for particles of diameter $d_c$ onto pre-existing aerosol and GR is the growth rate of the particles.
The coagulation sink for a particle of diameter $d_p$ is $CS(d_p) = \sum_j K(d_p, d_j) \cdot N_j$ , where $K(d_p, d_j)$ is the
coagulation coefficient for particles of diameter $d_p$ coagulating onto particles of diameter $d_j$. An assumption made
to derive Eq. 2 was that the coagulation coefficient for particles was proportional to the inverse of the square of the
particle diameter ( $\propto d_p^{-2}$). This is not always a sufficiently good approximation and the power dependency of the
coagulation coefficient can vary depending on the ambient particle size distribution which varies from one location
on the planet to another  (Kürten et al., 2015). For example, observations at Hyytiala in the Finnish boreal forest
(Dal Maso et al., 2005) reveal that the power law dependency of the coagulation sink with particle diameter is not
-2, it was in a range between -1.5 and -1.75. In a previous study (Lehtinen et al., 2007)  a new analytical expression
for $J_{3nm}$ was derived as shown in Equation 3.

$$J_{3nm} = J_{dc} exp \left( -\gamma \cdot d_c \cdot \frac{CS(d_c)}{GR} \right) \qquad …………….. (3)$$



Where $\gamma = \frac{1}{s+1}\left[\left(\frac{3}{dc}\right)^{s+1} - 1\right]$ $and$ $s = \frac{log\left(\frac{CS(3nm)}{CS(dc)}\right)}{log\left(\frac{3}{dc}\right)}$

We have incorporated this new expression into the model, and we show (Section 6) that this affects the concentration of smaller particles in the atmosphere by more correctly accounting for their losses due to coagulation.

*Coagulation sink.* The GLOMAP coagulation scheme (Jacobson et al., 1994) includes both inter-modal (collision
between particles that belong to different modes) and intra-modal (collision between particles in the same mode) coagulation. The estimation of the coagulation kernel has uncertainties in the effect of Van-der-Waals forces and charge on the particles (Nadykto and Yu, 2003). In this study we are focused only on the overall uncertainty of atmospheric processes, so we perturbed the model by scaling up the whole coagulation kernel by a factor of 5 to observe its impact on the model-observation comparison.

*Condensation Sink.* The two condensable species present in the model are $H_2SO_4$ (formed from the oxidation of $SO_2$) and Sec_org (formed from the oxidation of monoterpenes). The condensation sink refers to the rate at which these condensable gases condense onto aerosol particles in the atmosphere. It is conceivable that the presence of too much sulphuric acid in the atmosphere results in the formation of excess new particles, which could explain the bias in $N_{Total}$. Therefore, having a stronger condensation sink could help reduce the bias. The model also handles
the condensation of $H_2SO_4$ and Sec_org differently in that the sulphuric acid concentration is updated every microphysics time step (4min), while the Sec_org concentration is updated only on every chemistry time step (1hour). Since condensation in the atmosphere can happen on very short time scales, the Sec_org concentration may need to be updated at the end of every microphysics time step as well. We perform model runs after incorporating this change to the frequency at which Sec_org is updated, and also perform simulations where we
manually increase the condensation sink by a factor of 5 and 10 to see how sensitive the vertical profiles are to this perturbation (the condensation sink can also be indirectly affected by perturbations to other atmospheric processes). The motivation for increasing the condensation sink by large factors was to test the magnitude of the condensation sink required to reduce the large biases in $N_{Total}$. We only perturb the condensation sink directly, and not the $SO_2$ or particle number concentration, because perturbing the condensation sink is technically more straightforward.




## 5.2 DMS and Primary Marine Organic emissions

There is a significant uncertainty in gas phase DMS emission from the ocean, because the DMS emission fields are derived from a small set of ocean cruise measurements. Interpolation of this small data set (Kettle and Andreae, 2000; Lana et al., 2011) is used to obtain a global DMS emission field which is used by global models. This results

in a large uncertainty range in the DMS annual budget that lies between 17.6 – 34.4 Tg[S] (Lana et al., 2011). From past studies (McCoy et al., 2015; O'Dowd et al., 2004) we know that over marine regions, gas phase volatile organic compounds emitted from the ocean surface layer are a source of organic-enriched sea-spray aerosol. We also note that the DMS oxidation chemistry is also quite uncertain (Hoffmann et al., 2016; Veres et al., 2020) and this can lead to biases as well. Our default model version included an emission parametrization with the DMS field scaled

up by a factor of 1.7 to account for neglecting primary organic aerosol emissions in the model (Mulcahy et al., 2018). This simplified approach may not be realistic because scaling up DMS emissions will result in a larger production of $SO_2$ and $H_2SO_4$ via DMS and $SO_2$ oxidation. Since our goal is to reduce biases in $SO_2$ and particle number, we ran a simulation without the scale factor of 1.7. More recent versions of the model also include an emission parameterization to estimate the primary marine organic aerosol flux, which is significantly correlated to

the chlorophyll concentration (Gantt et al., 2012). Without removing the scale factor of DMS, we tested the sensitivity of aerosol number concentration to this parameterization by running model simulations with the primary marine organic emissions switched on, and also running simulations in which the emissions are scaled up by a factor of 5.

## 5.3 Cloud pH

Cloud droplet pH is an important parameter in the model because the aqueous phase oxidation of $SO_2$ by $O_3$ (to form sulphate) (Kreidenweis et al., 2003) is very sensitive to the pH of the cloud droplet. It is assumed in the model that this reaction occurs in all clouds, but the model only tracks the sulfate produced in shallow clouds, and not in deep convective clouds, since most of the sulphate formed would be scavenged from the atmosphere by precipitation in convective clouds, but not in non- or lightly-precipitating shallow clouds. The rate of this reaction

increases by a factor of $10^5$ for a pH change from 3 to 6 (Seinfeld and Pandis, 2016). Droplet pH is important because the consumption of $SO_2$ in a cloud droplet affects the mixing ratio of gas phase $SO_2$ available in the atmosphere, thereby reducing the gas phase concentration of $H_2SO_4$ (which can form particles). The cloud pH depends on the thermodynamic and kinetic processes in a changing cloud droplet distribution, which are not





explicitly simulated in our model; instead a constant cloud pH of 5 is assumed. This assumption could lead to

significant errors in regions of the planet where the pH is higher or lower than 5, owing to the regional variability in the amount of acidic and basic material present in the particles. Since we overestimate $SO_2$ compared to ATom observations, we performed perturbations by increasing the pH to 6 and 7 so as to lower the $SO_2$ and $N_{Total}$ bias. This parameter has also been identified in previous studies as one of the most important parameters for global CCN uncertainty (Lee et al., 2013).

**5.4 Scavenging of aerosol particles and gases**

The removal of aerosol particles and gases in convective clouds is an important atmospheric process that can control the vertical profiles of $N_{Total}$, $SO_2$ and condensation sink. Convection in the model is represented using a mass flux scheme (Gregory and Rowntree, 1990) which is responsible for the vertical transport of aerosol and gases. Understanding the effect of the removal mechanism for aerosol particles and gases during their vertical transport

is crucial in quantifying their vertical distribution. In the model, aerosol particles are scavenged using a convective plume scavenging scheme (Kipling et al., 2013), where scavenging coefficients for aerosol particles are assigned for each mode (denoted by the parameter 'rscav'). This convective plume scavenging scheme addresses, albeit crudely, biases that resulted from operator splitting between scavenging and convective transport and simulation of activation above cloud base, which were subsequently highlighted in other models (Yu et al., 2019). As a plume

rises through the atmosphere, the change in aerosol number and mass mixing ratios is dependent on the precipitation rate, convective updraught mass flux, mass mixing ratio of ice and liquid water, and the scavenging coefficients ('rscav') assigned to each mode. The nucleation mode is not scavenged and is assigned a scavenging coefficient of 0, the Aitken, accumulation and coarse modes are assigned scavenging coefficients of 0.5, 1 and 1 respectively. We assess the sensitivity of the model-observation comparison to perturbations in these values. These scavenging

coefficients used are consistent with convective cloud models which show that the aerosol in-cloud scavenging is close to the water scavenging efficiency (less than 1) (Flossmann and Wobrock, 2010).

We also scale up the convective rain scavenging rate for all gases (denoted by the parameter 'csca') by a factor of 10 and 20. These have higher uncertainty than aerosol scavenging coefficients because gas uptake into droplets and subsequent removal depends on gas solubility, temperature, ice formation (and gas retention during freezing), and

aqueous-phase chemistry (Yin et al., 2002).



## 5.5 Cloud erosion rate

The cloud erosion rate is an important tuning parameter (represented by UKESM parameter 'dbsdtbs') (Yoshioka et al., 2019) for the prognostic cloud fraction and prognostic condensate scheme (PC2) used in the model (Wilson et al., 2008). This parameter determines the rate at which un-resolved subgrid motions mix the clear and cloudy air, thereby removing liquid condensate, and it changes the cloud liquid fraction for shallow clouds. Changing this parameter should have an effect on $SO_2$ lifetime, as a result of its uptake into cloud droplets. Its effect on the fraction of cloud in each grid box will also change the amount of shortwave radiation received by Earth's surface which in turn can have feedback effects on aerosol processes. This parameter is usually tuned so that the outgoing shortwave radiation the model predicts matches observations. The default value of 'dbsdtbs' in the model is $1.5 \times 10^{-4}$. We perform two perturbation simulations with this value set to 0 and another with a value of 1e-3.

## 6. Results

The goal of the model one-at-a-time sensitivity tests is to understand the causes of biases in the model. Since we are interested in reducing the absolute magnitude of the biases we use the Normalised Mean Absolute Error Factor (NMAEF) (Yu et al., 2006) defined in Equation 4 instead of NMBF to characterise the bias. This new equation allows us to calculate the percentage change in model performance as the relative change in NMAEF of a model experiment with respect to the baseline version of UKESM as shown in Equation 5.

$$NMAEF = \begin{cases} \frac{\sum |M_i - O_i|}{\sum O_i}, & if \ \bar{M} \geq \bar{\bar{O}} \\ \frac{\sum |M_i - O_i|}{\sum M_i}, & if \ \bar{M} < \bar{\bar{O}} \end{cases} \dots\dots (4)$$

where $M_i$ represents model data, $O_i$ represents observations, $\bar{M}$ represents the model mean and $\bar{\bar{O}}$ represents the mean of the observations.

$$Percentage \ change \ in \ model \ performance = \left(1 - \frac{NMAEF_{simulation}}{NMAEF_{UKESM\_baseline}}\right) \times 100 \dots\dots (5)$$

The percentage change is zero when the sensitivity test has no effect on mean model bias, positive when there is an reduction in bias, and negative when the bias increases. A model that is in agreement with observations will have an NMAEF of zero and a percentage improvement of 100%. The vertical profiles for a few of the simulations



are shown in Figure 9. As we can see, different simulations have varying effects on the vertical profiles at different altitudes in the troposphere and we have therefore split our analysis to study model performance with altitude. The real boundary layer height varies with latitude, but for the purposes of this study we assume it is 1 km everywhere. Our results are similar for the boundary layer and lower troposphere, suggesting that our analysis is not sensitive to this assumed boundary layer height. In section 6.1 we look closely at the model's performance in the boundary

layer (which we define here as altitudes below 1 km) and lower troposphere (1 km < altitude < 4 km), and in Section 6.2 we study the mid (4 km < altitude < 8 km)  and upper troposphere (>8 km).

**6.1 Boundary layer and lower troposphere**

The performance for the different perturbation simulations in the boundary layer (altitude < 1 km) can be assessed from Figure 6. The NMAEF values for the simulations in the boundary layer are provided in Table 2a. The

percentage change in the bias of $N_{Total}$, $SO_2$ mixing ratio, and condensation sink from each of these perturbation simulations is calculated relative to the baseline version of UKESM and is represented by bar plots.

Firstly, we look at the model performance with respect to $N_{Total}$ in the altitude range 0-1 km where the model is biased low (Figure 6a). The baseline version of the model produces boundary layer $N_{Total}$ values that are negatively biased (NMAEF = 2.21). To reduce the bias in particle number concentration near the surface, the model

perturbation simulations (denoted as 'BL_nuc' and 'BL_nuc/10') that include a boundary layer nucleation mechanism show the best improvement in performance. 'BL_nuc' refers to the simulation that includes the Metzger boundary layer nucleation mechanism (Metzger et al 2010), and 'BL_nuc/10' refers to a simulation with the same nucleation mechanism but with the nucleation rate reduced by a factor of 10. Including this nucleation mechanism substantially improves model performance by 63% (NMAEF = 0.78) for 'BL_nuc' and 68% (NMAEF = 0.72) for

'BL_nuc/10'. This is an indication that the negative model bias in the boundary layer (Figure 3) could be explained by a missing boundary layer nucleation mechanism in the model, even though this mechanism depends on terrestrial emissions of shortlived organic compounds (typically not found in large concentrations over marine regions). A nucleation mechanism other than the Metzger mechanism (Metzger et al., 2010) which could be a scheme controlled by chemical species found in the marine boundary layer like methane sulfonic acid (MSA) (Pham et al.,

2005), iodine (Cuevas et al., 2018) or ammonia (Dunne et al., 2016) could help reduce model biases even more, but is not the focus of this work. All the other perturbation simulations either have no significant effect or decrease $N_{Total}$ model performance in the boundary layer. The perturbation simulations that stand out as performing the



poorest in the boundary layer are when we increase the pH (denoted by 'pH = 6' (NMAEF = 2.75) and 'pH = 7' (NMAEF = 2.94)), condensation sink (denoted by 'condsink*5' (NMAEF = 2.58) and 'condsink*10' (NMAEF = 2.89)) and scavenging of $SO_2$ ('csca*10' (NMAEF = 2.55) and 'csca*20' (NMAEF = 2.61)). These perturbations show (Figure 6a) an approximate decrease of 25% in $N_{Total}$ model performance.

Secondly, we look at the parameters that significantly improve the ability of the model to reproduce $SO_2$ mixing ratios in the boundary layer (Figure 6b) where the model is biased high (NMAEF = 2.09). Figure 6b shows that perturbations to cloud pH, DMS emissions (denoted as 'seadms=1.0'), convective rain scavenging rate (denoted by 'csca*10' and 'csca*20') and the cloud erosion rate (denoted by 'dbsdtbs=0') all improve model performance. The DMS emission perturbation, where we removed the artificial scaling factor of 1.7 that was used to compensate for the lack of primary marine organics, was also found to improve the model performance by 36% (NMAEF = 1.34). Increases in cloud pH from the default value of 5 to 6 or 7 (denoted in the figure as 'pH=6' and 'pH=7') improve the model by 34% (NMAEF = 1.39) and 48% (NMAEF = 1.09) respectively. In the atmosphere, a lower cloud pH is typically associated with polluted environments where particles are sulphate-rich, and higher cloud pH is associated with marine regions where particles are larger and contain carbonates from sea spray (Gurciullo and Pandis, 1997). Therefore, perturbations to cloud pH by increasing it to 6 or 7 are plausible explanations for the improved model skill since the observations are primarily over the remote ocean. Increasing the pH increases the rate of the reaction $SO_2 + O_3 \rightarrow SO_4^{2-}$ in a cloud droplet, thereby resulting in a larger consumption of aqueous $SO_2$. This drives more $SO_2$ from the gas phase to the aqueous phase, thereby reducing the gas phase $SO_2$ model bias. Increasing the pH can also compensate for the oxidation of $SO_2$ with $O_3$ on sea salt particles which is shown to be significant atmospheric process in marine regions (Korhonen et al., 2008). Furthermore, when the cloud erosion rate was set to zero (denoted by 'dbsdtbs_0'), it resulted in a model improvement of 25% (NMAEF = 1.56). A high value for dbsdtbs will cause more mixing of clear and dry air into clouds, thereby reducing the cloud liquid water content, cloud amount, and auto conversion of cloud droplets to raindrops. A low value of this parameter results in an increased lifetime for aerosol and precursor gases like $SO_2$.

Thirdly, we look at the parameters that most affect the model performance with respect to the prediction of the condensation sink (Figure 6c). The condensation sink in the boundary layer for the baseline version of the model has an NMAEF of 0.82. Simulations where we perturbed the boundary layer nucleation rate ('BL_nuc' and 'BL_nuc/10') and the primary marine organic emissions ('primmoc*5') showed a 15% (NMAEF = 0.69), 10% (NMAEF = 0.73) and 25% (NMAEF = 0.61) improvement in bias. This could be because the boundary layer is



lacking particles and including a new source of particles via boundary layer nucleation and emissions reduces the negative bias in the boundary layer (Figure 6c).

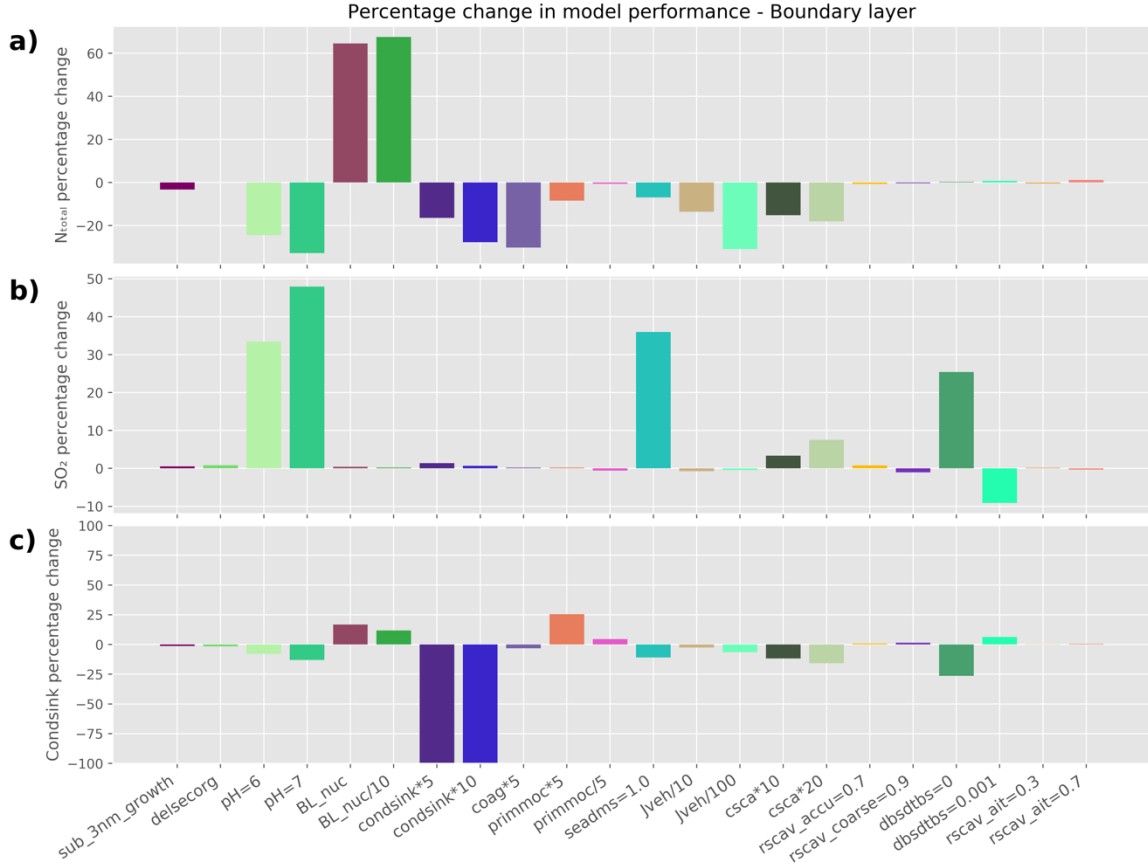

.

Figure 6: Percentage change in model performance for different perturbation simulations in the boundary layer (altitude < 1 km) with respect to, a) $N_{Total}$, b) $SO_2$, and c) condensation sink

The atmospheric processes that improve the skill of the model in the lower troposphere (between 1 km and 4 km) (Appendix A Figure A2) (NMAEF values are shown in Appendix A, Table A2) are the same as the boundary layer
with very slight differences in the magnitude of the percentage change in model performance.



## 6.2 Mid and Upper Troposphere

The model sensitivities in the upper troposphere are shown in Figure 7. Firstly, we assess $N_{Total}$ model performance for all the model simulations (Figure 7a). We observe that perturbations to several atmospheric processes help improve the model performance. Perturbations to the condensation sink, nucleation rate, sub 3nm growth, DMS emissions, gas scavenging rate, cloud erosion rate and cloud pH are found to have a significant effect on model performance. The range of parameter sensitivities is more diverse than in the boundary layer and the magnitudes are larger.

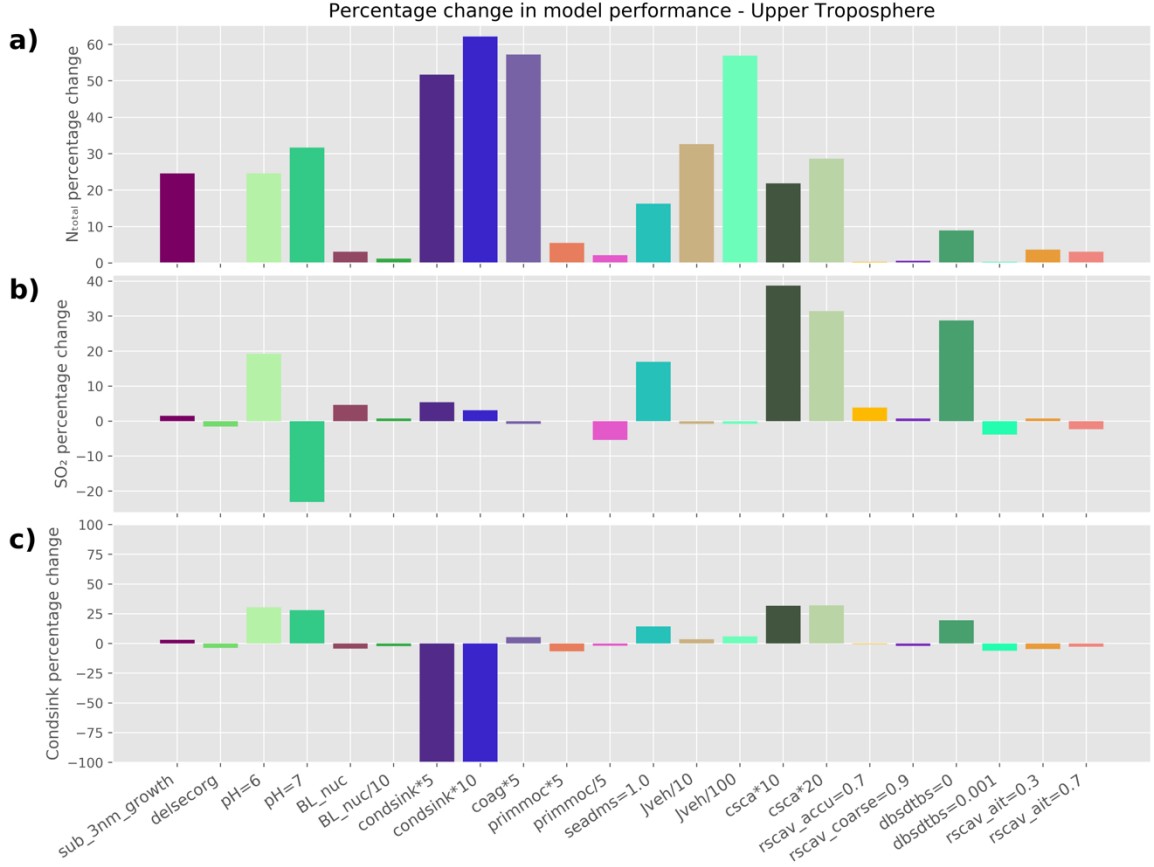





Figure 7: Percentage change in model performance for different perturbation simulations in the Upper Troposphere (>8 km ) with respect to, a) $N_{Total}$ , b) $SO_2$ , and c) condensation sink

First, we look at the model's performance with respect to $N_{Total}$. The baseline simulation produces $N_{Total}$ values that are biased high (NMAEF = 3.25) in the upper troposphere (Table 2b). The most improvement in model performance with respect to $N_{Total}$ (Figure 7a) was for the model simulations where we directly perturbed the condensation sink. These model runs were denoted as 'condsink*5' and 'condsink*10' and shows an improvement in performance by 51% (NMAEF = 1.57) and 62% (NMAEF= 1.23) respectively (Table 2b). This improvement in performance is because increasing the condensation sink will increase the rate at which $H_2SO_4$ is removed from the atmosphere via condensation onto particles. Therefore, increasing the condensation sink can help reduce the $H_2SO_4$ concentration and thus reduce the $N_{Total}$ bias.

Perturbations to nucleation rate where we reduced nucleation rate by a factor of 10 and 100 (denoted as 'Jveh/10' and 'Jveh/100') also improved the model by 32% (NMAEF = 2.19) and 56% (NMAEF = 1.4) respectively. This improvement in model performance by reducing nucleation rate is an indication that the source of the biases in $N_{Total}$ are mainly from small particles formed via nucleation. Model runs where we increase the convective gas scavenging rate (denoted as 'csca*10' and 'csca*20') by a factor of 10 and 20 results in a 21% (NMAEF = 2.54) and 28% (NMAEF = 2.32) improvement respectively. This scavenging rate simply scavenges the $SO_2$ from the atmosphere at a higher rate, which leaves less $SO_2$ to form $H_2SO_4$ via oxidation and therefore decreases $N_{Total}$. The cloud pH perturbation simulations show a 25% (NMAEF= 2.45) and 31% (NMAEF = 2.22) improvement for 'pH=6' and 'pH=7' respectively. Increasing cloud pH would increase the oxidation rate of $SO_2$ by ozone in cloud droplets (to form sulphate) thereby causing a reduction in the concentration of gaseous $H_2SO_4$. Incorporating the dependency of the coagulation sink on particle diameter (by using the (Lehtinen et al., 2007) parameterization denoted as 'sub_3nm_growth') reduces the positive bias in the model and improves the model by 24% (NMAEF = 2.45). This is because in the new expression (Lehtinen et al 2007) the coagulation sink for sub-3nm particles is greater than the previous assumption (Kerminen & Kulmula 2002).

Second, we analyse the model sensitivity and performance with respect to $SO_2$ (Figure 7b and Table 2). The baseline simulation produces $SO_2$ mixing ratios that are biased high (NMAEF = 1.3). The simulations that have the strongest effect on the biases are the perturbations to the DMS emissions ('seadms = 1.0'), cloud pH ('pH=6') and $SO_2$ scavenging rate ('csca*10' and 'csca*20'), they improve the model by 17% (NMAEF = 1.08) , 19% (NMAEF





= 1.05), 38% (NMAEF = 0.80) and 31% (NMAEF = 0.89) respectively (Table 2). The large $SO_2$ over-prediction
by the model in the tropical upper-troposphere (NMAEF = 1.3) is corrected by the perturbations where the $SO_2$ in
the atmosphere is removed by scavenging ('csca*10' and 'csca*20'), by reduction in DMS emissions
('seadms=1.0') or by reduction in the $SO_2$ mixing ratio as a result of increasing the cloud droplet pH. However, the
simulation with cloud pH set to 7 results in too much $SO_2$ being removed by lower level clouds, leaving less
available $SO_2$ to be convected to the upper troposphere causing a large negative bias (NMAEF = 1.6).

Third, we look at the model performance with respect to the condensation sink (Figure 7c) where the model is
biased with NMAEF = 0.61. The perturbations; cloud pH ('pH =6' and 'pH=7'), convective gas scavenging rate
('csca*10' and 'csca*20'), cloud erosion rate ('dbsdtbs=0') and DMS emissions ('seadms =1.0') all improve model
performance by 15-30%. Increasing the pH of a cloud drop enhances $SO_2$ aqueous phase chemistry in low level
clouds to form sulphate, which partitions sulfur to the aqueous phase and increases wet removal, leaving less $SO_2$
to be convected upward to higher altitudes. This also results in a reduction in the concentration of larger particles
being transported by convection to higher altitudes, thereby reducing the condensation sink to match better with
observations. Similarly, reduction in cloud erosion rate will result in greater uptake of $SO_2$ on cloud droplets to
form sulphate, thereby increasing aerosol mass and increasing the amount of scavenged larger particles. The other
perturbations, where we indirectly influenced the $SO_2$ mixing ratio in the atmosphere by reducing the DMS
emissions and $SO_2$ scavenging, also reduce the positive bias in the model condensation sink by reducing the $SO_2$
available to form sulphate.

The atmospheric processes that are of significance to model performance with respect to $N_{Total}$ and condensation
sink in the mid troposphere are similar to the upper troposphere, with decreases in the magnitude of model
performance (Figure A3, Appendix) relative to the upper troposphere. This indicates that the atmospheric processes
that have been identified are of more importance at higher altitudes. However, for the model performance with
respect to $SO_2$ in the mid troposphere shows more similarity with the lower troposphere (Figure A2, Appendix).

**7. Model performance: A three-way comparison**

**7.1. Effect of perturbations on multiple variables**

The main reason for analysing $N_{Total}$, $SO_2$ and condensation sink model performance simultaneously is to make
sure that performing one-at-a-time sensitivity tests to assess model performance leads to a consistent result.



Improving only one of these quantities in comparison with observations can lead to a misleading impression that overall model performance has improved. Analysing $N_{Total}$, $SO_2$ and condensation sink simultaneously helps reduce the probability of getting the right answer for the wrong reasons. We find that different atmospheric processes affect the vertical profile of $N_{Total}$, $SO_2$ and condensation sink to varying degrees.

Firstly, we analyse the boundary layer (<1 km) and lower troposphere (1-4 km). In section 6.1 we identified the atmospheric processes that are important for the boundary layer and how they affected model performance with respect to $N_{Total}$, $SO_2$ and condensation sink independently. Here we look at which simulations perform the best when comparing these variables simultaneously. Table 2 shows the NMAEF in the boundary layer and upper troposphere for all of the simulations. The NMAEF values for the baseline simulation are highlighted in yellow,

the green boxes represent NMAEF values for the simulations that have the same or lower biases than the baseline simulation, and the orange boxes represent those simulations that have higher biases than the baseline simulation. The results show that the model simulations where we perturbed the cloud pH, DMS emissions, convective gas scavenging rate and cloud erosion rate all significantly reduce biases with respect to $SO_2$ but make the model perform worse with respect to $N_{Total}$ and the condensation sink. In Table 2, the blue dotted boxes highlight the

simulations for which the biases with respect to $N_{Total}$, $SO_2$ and condensation sink are less than or equal to the baseline simulation. The only model simulation that improved $N_{Total}$, $SO_2$ and condensation skill simultaneously was when we included boundary layer nucleation ('BL_nuc' and 'BL_nuc/10'). Including a boundary layer nucleation scheme adds a new source of particles which helps reduce the negative bias the model shows in the boundary layer.



| a) | NMAEF for model simulations in the Boundary layer | | |
|---|---|---|---|
| Model perturbation | $N_{Total}$ | $SO_2$ | Condensation sink |
| Baseline | 2.21 | 2.09 | 0.82 |
| sub_3nm_growth | 2.28 | 2.08 | 0.83 |
| delsecorg | 2.21 | 2.07 | 0.84 |
| pH=6 | 2.75 | 1.39 | 0.89 |
| pH=7 | 2.94 | 1.09 | 0.93 |
| BL_nuc | 0.78 | 2.08 | 0.69 |
| BL_nuc/10 | 0.72 | 2.09 | 0.73 |
| condsink*5 | 2.58 | 2.06 | 2.46 |
| condsink*10 | 2.82 | 2.08 | 5.55 |
| coag*5 | 2.88 | 2.09 | 0.85 |
| primmoc*5 | 2.40 | 2.09 | 0.61 |
| primmoc | 2.22 | 2.10 | 0.79 |
| seadms=1.0 | 2.36 | 1.34 | 0.91 |
| Jveh/10 | 2.51 | 2.11 | 0.85 |
| Jveh/100 | 2.89 | 2.10 | 0.88 |
| csca*10 | 2.55 | 2.02 | 0.92 |
| csca*20 | 2.61 | 1.93 | 0.95 |
| rscav_accu=0.7 | 2.23 | 2.07 | 0.82 |
| rscav_coarse=0.9 | 2.22 | 2.11 | 0.81 |
| dbsdtbs=0 | 2.21 | 1.56 | 1.04 |
| dbsdtbs=0.001 | 2.19 | 2.28 | 0.77 |
| rscav_ait=0.3 | 2.22 | 2.09 | 0.82 |
| rscav_ait=0.7 | 2.19 | 2.10 | 0.82 |

| b) | NMAEF for model simulations in the upper troposphere | | |
|---|---|---|---|
| Model perturbation | $N_{Total}$ | $SO_2$ | Condensation sink |
| Baseline | 3.25 | 1.30 | 0.61 |
| sub_3nm_growth | 2.45 | 1.28 | 0.59 |
| delsecorg | 3.25 | 1.32 | 0.63 |
| pH=6 | 2.45 | 1.05 | 0.42 |
| pH=7 | 2.22 | 1.60 | 0.44 |
| BL_nuc | 3.15 | 1.24 | 0.63 |
| BL_nuc/10 | 3.21 | 1.29 | 0.62 |
| condsink*5 | 1.57 | 1.23 | 5.80 |
| condsink*10 | 1.23 | 1.26 | 12.10 |
| coag*5 | 1.39 | 1.31 | 0.57 |
| primmoc*5 | 3.07 | 1.30 | 0.65 |
| primmoc | 3.18 | 1.37 | 0.62 |
| seadms=1.0 | 2.72 | 1.08 | 0.52 |
| Jveh/10 | 2.19 | 1.30 | 0.58 |
| Jveh/100 | 1.40 | 1.30 | 0.57 |
| csca*10 | 2.54 | 0.80 | 0.41 |
| csca*20 | 2.32 | 0.89 | 0.41 |
| rscav_accu=0.7 | 3.26 | 1.25 | 0.61 |
| rscav_coarse=0.9 | 3.26 | 1.29 | 0.62 |
| dbsdtbs=0 | 2.96 | 0.93 | 0.49 |
| dbsdtbs=0.001 | 3.24 | 1.35 | 0.64 |
| rscav_ait=0.3 | 3.19 | 1.29 | 0.63 |
| rscav_ait=0.7 | 3.27 | 1.33 | 0.62 |


Table 2: Normalised mean absolute error factor (NMAEF) with respect to $N_{Total}$, $SO_2$ and condensation sink for different model simulations. NMAEF values for the baseline simulation are highlighted in yellow. NMAEF values that are less than or equal to the baseline simulation are highlighted in green. NMAEF values that are greater than the baseline simulation are highlighted in orange. The dotted blue box indicates the model simulation for which

NMAEF values for $N_{Total}$, $SO_2$ and condensation sink are less than the baseline simulation simultaneously; a) boundary layer (below 1km) and b) upper troposphere (>8km)

In the upper troposphere (Table 2b), several simulations improve $N_{Total}$ model performance. The positive model bias in $N_{Total}$ is significantly reduced by perturbations to the sub 3 nm growth, cloud pH, condensation sink, coagulation sink, primary marine organic emissions, DMS emissions, nucleation rate, and $SO_2$ gas scavenging rate.

Direct perturbations to the condensation sink, although they improve $N_{Total}$ model skill significantly, worsen the model performance with respect to the condensation sink (NMAEF = 12.1 for 'condsink*10' simulation). Thus, from Table 2b, the blue dotted boxes indicate the simulations for which the model biases for $N_{Total}$, $SO_2$ and condensation sink are less than (or equal to) the baseline version of the model simultaneously.





Figure 8: Diagram to represent of the $N_{Total}$, $SO_2$ and condensation sink biases (in the boundary layer and upper troposphere) for the one at time sensitivity tests: sub 3nm growth, Cloud pH = 6, scaling down DMS emissions,





boundary layer nucleation/10. The blue, green and black legs of the diagram represent the $N_{Total}$, $SO_2$ and condensation sink bias respectively. The yellow and pink bars represent the biases in the boundary layer and upper troposphere normalised with respect to the baseline simulation.

We see this simultaneous reduction of biases in the mid (Table A2 appendix) and upper troposphere for simulations where we perturbed sub 3nm growth, cloud pH, DMS emissions, nucleation rate, $SO_2$ gas scavenging rate and cloud erosion rate. The one main difference between the simulations in the mid and upper troposphere is that the perturbation to cloud pH (pH =7) improves overall model performance in the mid-troposphere but not in the upper troposphere. At pH = 7 the model in the upper troposphere also shows a larger $SO_2$ bias (NMAEF = 1.6) than the
baseline (NMAEF = 1.3).





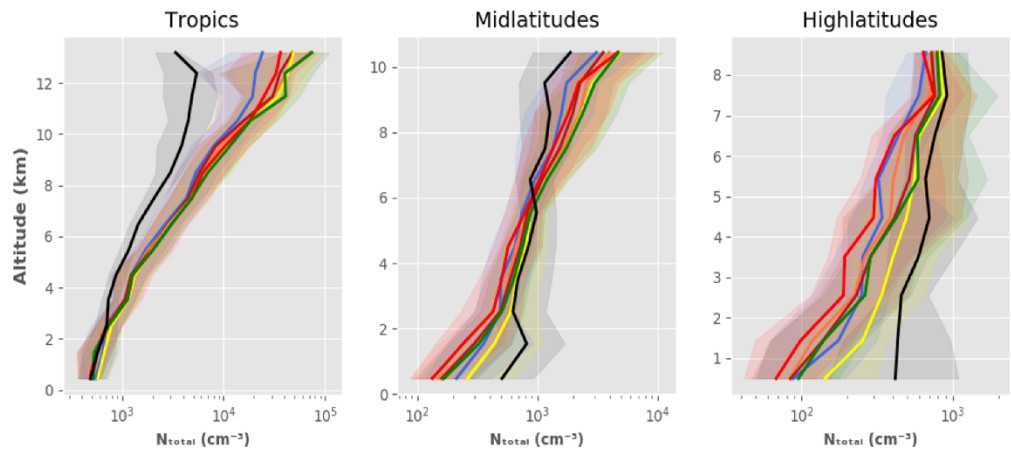

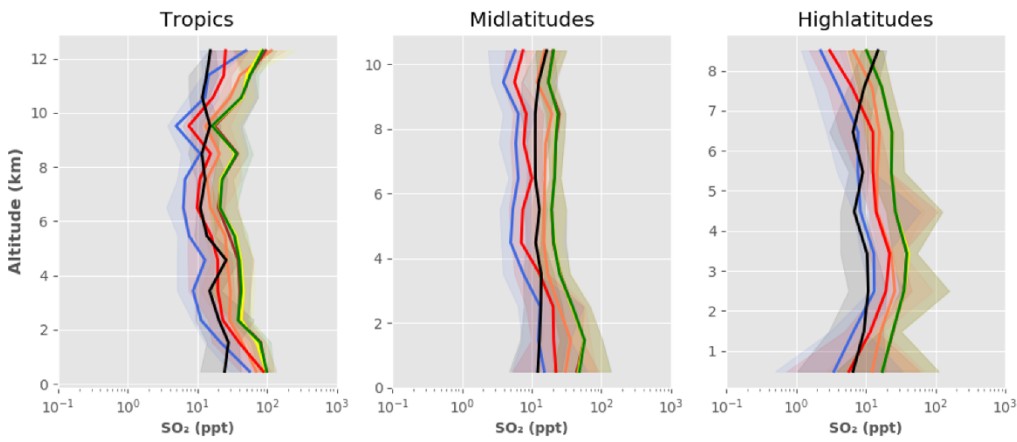

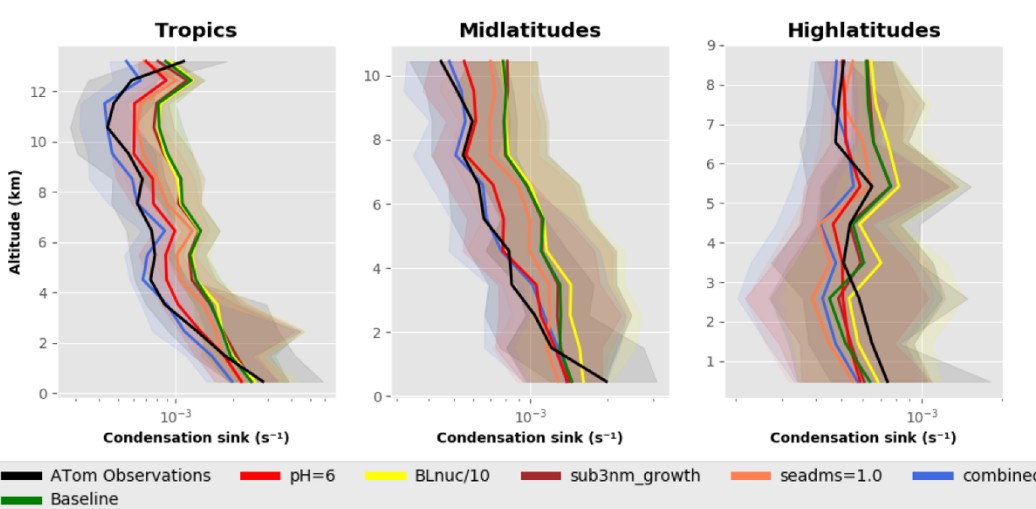





Figure 9: The vertical profile of a) $N_{Total}$, b) $SO_2$ and c) condensation sink for different model experiments that were found to have the most influence on model performance. The vertical profiles of observation data, the baseline simulation and perturbation simulations of cloud pH, boundary later nucleation, sub-3 nm growth, scaled-down DMS emissions, and the combined simulation are shown and categorised into three regions of the earth: the tropics (25°N-25°S), midlatitudes (25°N-60°N and 25°S-60°S), and high latitudes (60°N-90°N and 60°S-90°S).

We show the combined model bias for a select few sensitivity tests in the boundary layer and upper troposphere using a bar diagram (Figure 8). In this presentation, the blue, green and black bars represent the normalised bias in $N_{Total}$, $SO_2$ and condensation sink for the baseline simulation. The yellow and pink bars represent the corresponding biases in the boundary layer and upper troposphere for any given sensitivity test (normalised with respect to the baseline simulation). If the length of the blue, green or black bars is greater than the length of the corresponding yellow and pink bar, then the bias in the sensitivity test is less than the baseline simulation. The vertical profiles for the simulations used in Figure 8 are shown in Figure 9. Simulations where we perturbed sub-3 nm growth, pH = 6, DMS scaling, and boundary layer nucleation/10 showed a reduction in biases and in some cases increased biases negligibly. The boundary layer nucleation simulation (BL_nuc/10) reduces biases in the boundary layer $N_{Total}$ by ~67% without affecting the upper tropospheric $N_{Total}$ bias. This simulation does not have any effect on the $SO_2$ mixing ratio but does reduce the condensation sink bias in the boundary layer by ~11% and shows a negligible change in bias (~2%) in the upper troposphere. Changing the pH to 6 causes a slight degradation in the model's $N_{Total}$ and condensation sink (increase in bias by ~24% and ~8%) in the boundary layer and improved the $SO_2$ by 33%. However, in the upper troposphere perturbations to pH has a positive effect on model performance against observations. The 'sub_3nm_growth' simulation improves the upper tropospheric $N_{Total}$ bias by ~24% without significantly affecting other parameters. Removing the scaling factor for DMS emission helps improve the upper tropospheric $N_{Total}$, $SO_2$, and condensation sink bias by 16%, 17% and 14% respectively. It also reduces the boundary layer $SO_2$ bias by 35% and shows a small increase of 6% and 10% in the $N_{Total}$ and condensation sink bias respectively. Thus, we have identified the perturbation simulations; 'BL_nuc/10', 'pH = 6', 'Seadms = 1.0' and 'sub_3nm_growth' as the simulations that help reduce model biases in most cases across $N_{Total}$, $SO_2$ and condensation in the boundary layer and upper troposphere. These perturbations are well-motivated in that they improve the physical basis of the model and can be looked at more closely when developing future versions of UKESM.



## 7.2. Effect of combined perturbations on multiple variables

We performed one simulation incorporating the 4 perturbations (BL_nuc/10, pH = 6, 'Seadms = 1.0' and 'sub_3nm_growth') discussed in section 7.1 simultaneously (bottom row in Figure 8) to assess model performance. For $N_{Total}$, the model's boundary layer and upper tropospheric performance is improved (NMAEF reduced by 24%

and 54% respectively). The positive $SO_2$ bias improves by 54% in the boundary layer but showed a slight degradation of 10% in the upper troposphere. The positive condensation sink bias shows a negligible increase of 4% in the boundary layer and a 29% decrease in the upper troposphere. From Figure 9, the $SO_2$ profile for the combined simulation shows better agreement with observations in the tropics and high latitudes and shows a small negative bias in the midlatitude free troposphere. The condensation sink profile of the combined simulation does

show a much better agreement with the observations in tropics, midlatitudes and high latitudes. The combined simulation also shows a substantial reduction in the upper tropospheric $N_{Total}$ bias in the tropics and midlatitudes but the large negative bias in the high latitudes remains, and at high altitudes in the high latitude regions, it is exacerbated. In the boundary layer, the combined simulation shows a small improvement in the midlatitudes but otherwise performs similar to the baseline simulation.

In the tropical free troposphere, the fact that the $SO_2$ and condensation sink for the combined simulation agree very well with observations and $N_{Total}$ is still overpredicted suggests a missing loss process for nucleation mode particles in the upper troposphere, or a bias in the downward transport of these particles to lower altitudes. The biases in $N_{Total}$ in the high-latitude and mid-latitude boundary layer for the combined simulation could be because of a missing source of small particles from a marine nucleation mechanism which is not included in the model, for

example involving iodine or methane sulfonic acid (Baccarini et al., 2018; Hodshire et al., 2019). Even though simulations with the Metzger boundary layer nucleation scheme (Metzger et al., 2010) helped reduce this bias, this nucleation scheme is primarily dependent on the concentration of organic vapors from terrestrial sources, which are low over marine regions. The biases in the boundary layer high latitudes could also be due to uncertainties associated with the sea spray parametrisation in the model (Regayre et al., 2020).

To summarise, our new combined simulation performs significantly better than the baseline model we started with for all three variables, $N_{Total}$, $SO_2$ and condensation sink. However, we were still unable to reproduce observations of $N_{Total}$ in the tropical free troposphere, the mid-latitude boundary layer, and the high latitudes with the well-





motivated adjustments we applied. Clearly structural errors in the model remain: this study motivates future model developments to address the biases and indicates where the developments should be focused.

**8. Discussion and Conclusions**

We have evaluated the vertical profile of $N_{Total}$, $SO_2$ and condensation sink from UKESM against ATom aircraft measurements. The model captured the trends in the vertical profiles. Quantitatively, the model reproduced the vertical profile of condensation sink moderately well but shows higher biases in the $N_{Total}$ and $SO_2$ vertical profile. We performed model simulations to help understand which atmospheric processes influence the model skill and 720 thereby help match the model's prediction of $N_{Total}$, $SO_2$, and condensation sink simultaneously with observations. We found that different atmospheric processes have a varying impact on model skill with altitude.

In the boundary layer and lower troposphere, the model showed negative biases in $N_{Total}$ (up to a factor of 3) and positive biases in $SO_2$ (up to a factor of 6) with moderate positive/negative model biases in the condensation sink (within a factor of 2). We found that simulations with boundary layer nucleation included were the only simulations 725 that reduced the biases in $N_{Total}$ and condensation sink in the boundary layer simultaneously with negligible changes to the $SO_2$ mixing ratio.

In the middle and upper troposphere, the largest biases were again observed in $N_{Total}$ (positive biases up to a factor of 15) and $SO_2$ (positive biases up to a factor of 6), with the model's condensation sink showing modest positive/ negative biases (within a factor of 2). However, in contrast to lower altitudes, we found that adjustment of several 730 atmospheric processes improved overall model performance. From our one-at-a-time sensitivity tests we found that simulations with perturbations to the sub-3 nm growth, cloud pH, DMS emissions, nucleation rate, gas scavenging rate and cloud erosion rate all help reduce model biases in $N_{Total}$, $SO_2$, and condensation sink simultaneously at higher altitudes.

Simulations where we increased the condensation sink by a factor of 10 or reduced the nucleation rate by a factor 735 of 100 also substantially improved the model's $N_{Total}$ profile in the tropical upper troposphere. However, while useful to understand the sensitivity, artificial adjustment of the condensation sink is unrealistic because the model shows only a factor of 2 bias compared to observations. Substantial reduction of the nucleation rate was also explored as this is the main source of particles in the cold upper troposphere. However, the default nucleation rate (Vehkamäki et al., 2002) has been shown to be reasonably accurate or even underestimated for a given sulphuric





acid concentration, temperature and humidity (Määttänen et al., 2018). If the effective nucleation rate in the model is indeed too high by a factor of 100, then this may instead suggest a structural deficiency in the way nucleation is implemented in the model, which we discuss below. Any adjustment of the nucleation rate itself is not supported by our current understanding of the rate of nucleation under upper tropospheric conditions.

Though there are differences in the importance of certain atmospheric processes over others at low and high

altitudes, we have identified a few well-motivated changes that help reduce the bias in the boundary layer and upper troposphere. From our analysis we can suggest the following,

1. Including a boundary layer nucleation scheme helps reduce model biases at lower altitudes without causing large changes in biases in the upper troposphere.
2. Changing the value of cloud pH from 5 to 6 produces a significant improvement in model performance in

the mid and upper troposphere. However, this change does result in a slight degradation of the model's $N_{Total}$ profile at lower altitudes.
3. Improvements to the model's microphysics by updating the parameterization of nuclei growth (Kerminen and Kulmala, 2002) to include a corrected dependency of coagulation sink on particle diameter (Lehtinen et al., 2007) improves upper tropospheric model performance without significant degradation of the model

at lower altitudes.
4. Removing the scaling factor for DMS emissions also helps reduce the positive biases in $SO_2$ both in the boundary layer and upper troposphere. This simulation does however increase the biases in $N_{Total}$ and condensation sink in the boundary layer.

We performed a simulation with these four perturbations included simultaneously and found the model's

performance in the boundary layer and upper troposphere improved simultaneously. The combined simulation's $SO_2$ and condensation sink profiles agree very well with observations and perform much better than the baseline simulation. However, the $N_{Total}$ profile for the combined simulation in the tropics and high-latitudes, while performing better than the baseline simulation, still has significant biases when compared to observations. The fact that this adjusted simulation reduces the $N_{Total}$ bias, but does not completely eliminate it, will help us identify the

possible deficiencies of the model in future work. The absence of a scavenging mechanism for nucleation mode particles (for example on cirrus clouds) or uncertainties in the downward transport of particles could explain the reason for the $N_{Total}$ positive bias in the upper troposphere-tropics. The negative bias in the boundary layer $N_{Total}$





could be explained by uncertainties associated with the sea spray parametrisation or the absence of a nucleation scheme involving gaseous precursors found in the marine environment. Thus, in this work, we have identified several atmospheric processes and parameters in UKESM that are key to the skilful simulation of $SO_2$ mixing ratio, condensation sink and $N_{Total}$ simultaneously, although we reached a limit in how much the $N_{Total}$ can be improved upon with the current set of simulations. These perturbations shed light on the influence of different atmospheric processes on aerosol number concentration and motivate further development of parameterizations in the model. Our work will also help inform future perturbed parameter ensemble studies designed to analyse and constrain the effect of a combination of parameters on model skill.

## 9. Data availability

All model and observation data used in this study can be accessed from https://doi.org/10.5281/zenodo.4088640. The observations from the ATom campaign can be also be obtained from (Wofsy et al., 2018) or https://espoarchive.nasa.gov/archive/browse/atom.

## 10. Author Contributions

AR, HG and KC designed the idea for this study. AR performed all model simulations and analysis with guidance from HG and KC. CW, AK, ARo and CB were responsible for the data from the ATom campaign used in this study. HG and LA helped identify and resolve the bug in the model code discussed in the article. KP provided scientific guidance and infrastructure support. AR wrote the manuscript with help from HG and KC, and contributions from all co-authors.

## 11. Acknowledgements

We thank the ATom science team and the NASA DC-8 flight crew for their contributions to the ATom dataset. We also thank the UK Met office for giving us access to their state-of-the-art supercomputing facilities, and all the people responsible for the development of UKESM.

## 12. Financial support

This research has been supported by Marie Sklodowska Curie no. 764991 "CLOUDMOTION"



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



# Appendix A









**Figure A1:** a) The vertical profile of the total particle number concentration, b) The vertical profile of $SO_2$ mixing ratio and c) The vertical profile of the condensation sink. The Vertical profiles are provided for the Northern and southern Midlatitudes ($25^{o}$N-$60^{0}$N and $25^{o}$S-$60^{o}$S) as well as the northern and southern highlatitudes ($60^{o}$N-$90^{o}$N and $60^{o}$S-$90^{o}$S). The bold line represents the median and the shaded region represents the corresponding interquartile range ($25^{th}$ and $75^{th}$ percentile) in a 1km altitude bin.



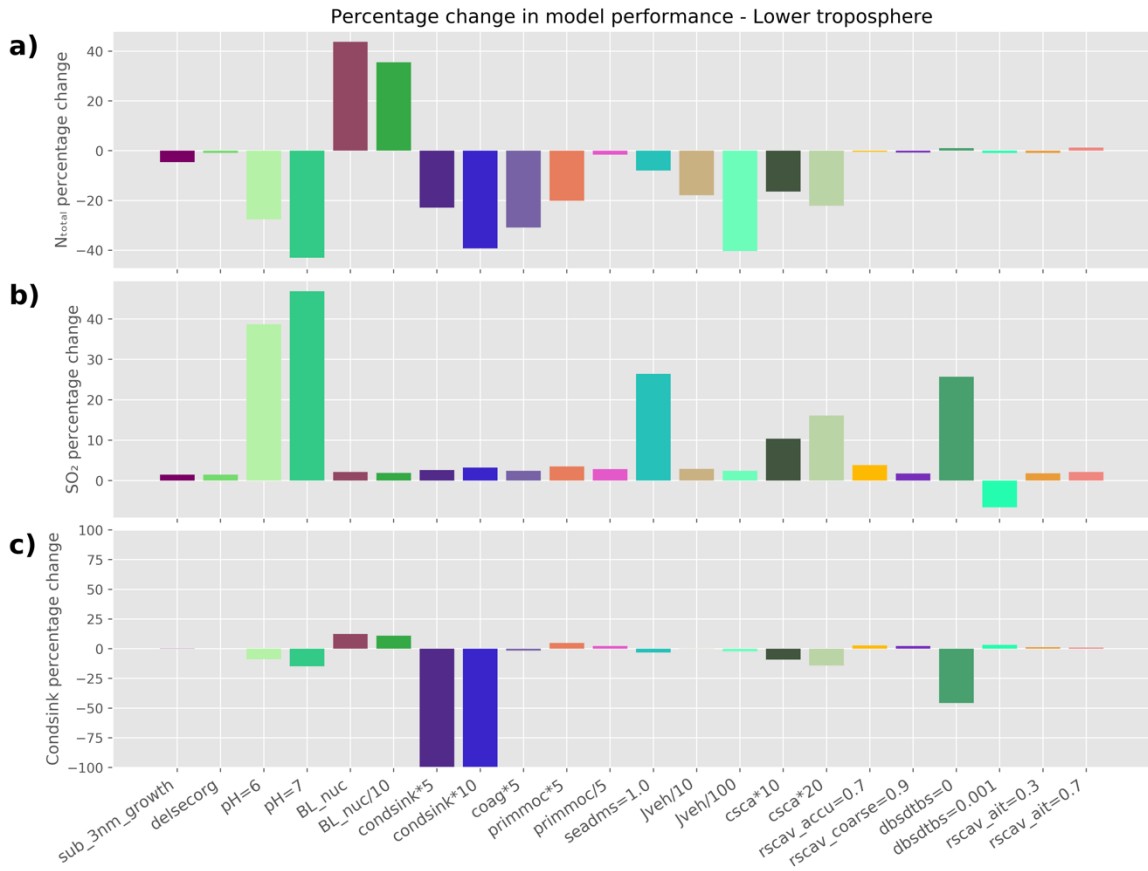

Figure A2: Percentage change in model performance for the different perturbation experiments in the Lower
Troposphere (1km < altitude < 4km) with respect to, a) $N_{Total}$, b) $SO_2$, and c) condensation sink



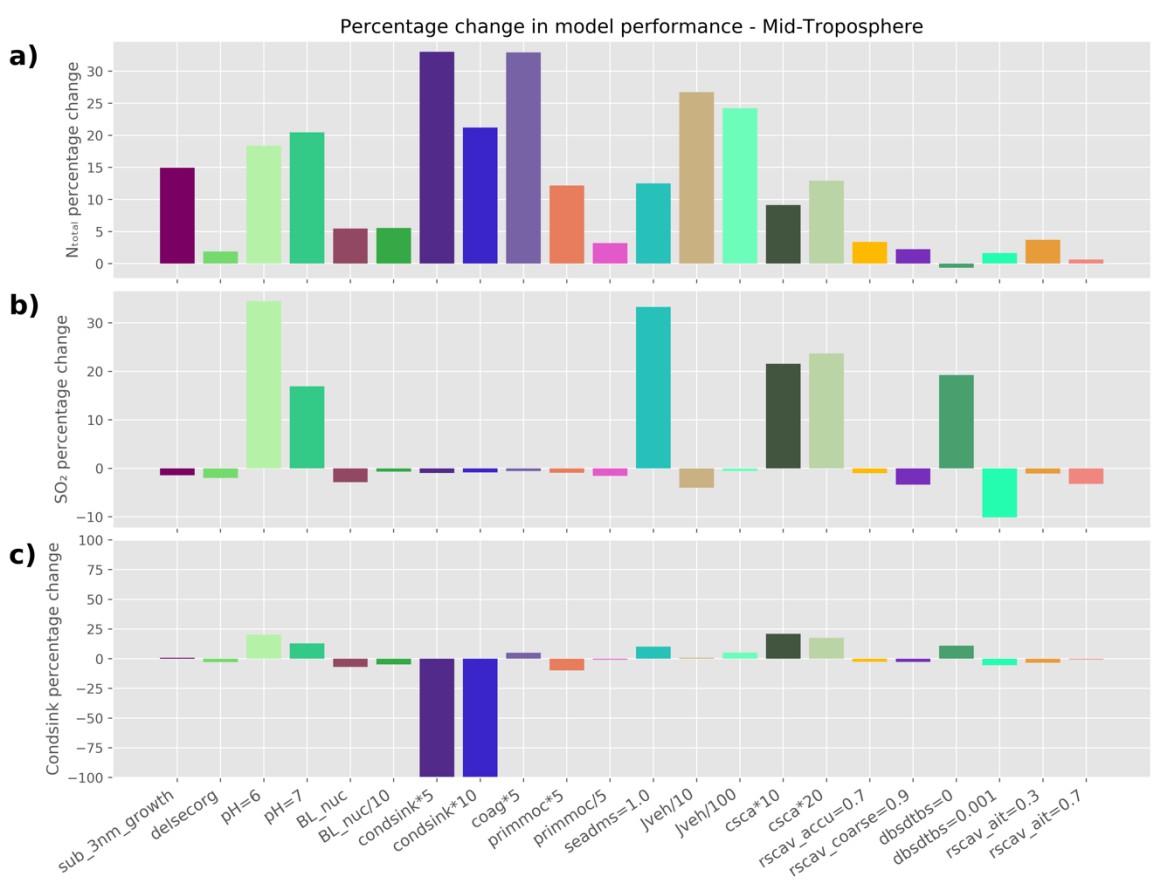

Figure A3: Percentage change in model performance for the different perturbation experiments in the Mid Troposphere (4km < altitude < 8km) with respect to, a) $N_{Total}$, b) $SO_2$, and c) condensation sink





Table A1: The different aerosol size modes in UKESM along with their size ranges, mode standard deviation and
aerosol species in each mode. The species are organic matter (OM), sulphate (SO₄), BC (black carbon) and sea salt.
Dust is treated separately as described in the text.

| Aerosol Mode | Geometric mean diameter $\bar{d}$(nm) | Mode standard deviation | Species |
|---|---|---|---|
| Nucleation Soluble | $\bar{d} < 10$ nm | 1.59 | OM, SO₄ |
| Aitken Soluble | 10 nm $< \bar{d} < 100$ nm | 1.59 | OM, SO₄, BC |
| Accumulation Soluble | 100 nm $< \bar{d} < 500$ nm | 1.40 | OM, SO₄, BC, Sea salt |
| Coarse Soluble | 500 nm $< \bar{d} < 10000$ nm | 2.00 | OM, SO₄, BC, Sea salt |
| Aitken Insoluble | 10 nm $< \bar{d} < 100$ nm | 1.59 | OM, BC |



Table A2: Normalised mean absolute error factor (NMAEF) wSith respect to $N_{Total}$, $SO_2$ and condensation sink for
different model simulations. NMAEF values for the baseline simulation is highlighted in yellow. NMAEF values
that are less than (or equal to) the baseline simulation are highlighted in green. NMAEF values that are greater than
the baseline simulation are highlighted in orange. The dotted blue box indicates the model simulation for which
NMAEF values for $N_{Total}$, $SO_2$ and condensation sink are less than (or equal to) the baseline simulation
simultaneously; a) lower troposphere (between 1km and 4km) and b) mid troposphere (between 4km and 8km)

a)

| NMAEF for model simulations in the lower troposphere | | | |
|---|---|---|---|
| Model perturbation | $N_{Total}$ | $SO_2$ | Condensation sink |
| Baseline | 1.21 | 2.13 | 0.71 |
| sub_3nm_growth | 1.27 | 2.10 | 0.71 |
| delsecorg | 1.23 | 2.10 | 0.71 |
| pH=6 | 1.55 | 1.31 | 0.78 |
| pH=7 | 1.74 | 1.13 | 0.82 |
| BL_nuc | 0.68 | 2.09 | 0.63 |
| BL_nuc/10 | 0.78 | 2.09 | 0.64 |
| condsink*5 | 1.49 | 2.08 | 3.18 |
| condsink*10 | 1.69 | 2.06 | 7.01 |
| coag*5 | 1.59 | 2.08 | 0.73 |
| primmoc*5 | 1.46 | 2.06 | 0.68 |
| primmoc | 1.23 | 2.07 | 0.70 |
| seadms=1.0 | 1.31 | 1.57 | 0.74 |
| Jveh/10 | 1.43 | 2.07 | 0.72 |
| Jveh/100 | 1.70 | 2.08 | 0.73 |
| csca*10 | 1.41 | 1.91 | 0.78 |
| csca*20 | 1.48 | 1.79 | 0.82 |
| rscav_accu=0.7 | 1.22 | 2.05 | 0.70 |
| rscav_coarse=0.9 | 1.22 | 2.09 | 0.70 |
| dbsdtbs=0 | 1.20 | 1.58 | 1.04 |
| dbsdtbs=0.001 | 1.23 | 2.27 | 0.69 |
| rscav_ait=0.3 | 1.23 | 2.09 | 0.71 |
| rscav_ait=0.7 | 1.20 | 2.08 | 0.72 |

b)

| NMAEF for model simulations in the mid troposphere | | | |
|---|---|---|---|
| Model perturbation | $N_{Total}$ | $SO_2$ | Condensation sink |
| Baseline | 1.15 | 1.27 | 0.58 |
| sub_3nm_growth | 0.97 | 1.27 | 0.57 |
| delsecorg | 1.12 | 1.29 | 0.59 |
| pH=6 | 0.93 | 0.83 | 0.46 |
| pH=7 | 0.91 | 1.05 | 0.50 |
| BL_nuc | 1.08 | 1.30 | 0.62 |
| BL_nuc/10 | 1.08 | 1.27 | 0.61 |
| condsink*5 | 0.77 | 1.28 | 5.08 |
| condsink*10 | 0.90 | 1.28 | 10.76 |
| coag*5 | 0.77 | 1.28 | 0.55 |
| primmoc*5 | 1.01 | 1.28 | 0.63 |
| primmoc | 1.11 | 1.28 | 0.58 |
| seadms=1.0 | 1.00 | 0.84 | 0.52 |
| Jveh/10 | 0.84 | 1.32 | 0.57 |
| Jveh/100 | 0.87 | 1.27 | 0.55 |
| csca*10 | 1.04 | 0.99 | 0.46 |
| csca*20 | 1.00 | 0.97 | 0.48 |
| rscav_accu=0.7 | 1.11 | 1.28 | 0.59 |
| rscav_coarse=0.9 | 1.12 | 1.31 | 0.59 |
| dbsdtbs=0 | 1.15 | 1.02 | 0.51 |
| dbsdtbs=0.001 | 1.13 | 1.39 | 0.61 |
| rscav_ait=0.3 | 1.10 | 1.28 | 0.60 |
| rscav_ait=0.7 | 1.14 | 1.31 | 0.58 |