# Peer review of "Constraints on global aerosol number concentration, SO2 and condensation sink in UKESM1 using ATom measurements"

_Atmospheric Chemistry and Physics, 2020_

## Referee Comment (RC1) · Anonymous Referee #1 · 14 Dec 2020

This paper focuses on simultaneously documenting and improving model ability to accurately simulate the vertical distribution of three interlinked atmospheric variables (SO2 mixing ratio, particle number concentration and condensation sink). The authors perform an extensive set of parameter perturbation simulations and compare the results to measurements from the ATom flights, which provide a unique dataset for model validation with a broad geographical coverage. The vertical distribution of particles has been the focus of much research over a long time, with persisting model-measurement discrepancies. This paper focuses on previously less well studied variables and their relationship, hence make an important contribution to the scientific knowledge base. Based on the results from the sensitivity simulations, some suggestions for general

improvements are provided. Overall, the paper is well-written and organized, but it is quite lengthy, and I think an outline of the sections at the beginning would be helpful. In addition, I have some comments, which are mainly minor ones for clarification.

General: - I'm puzzled by the use of model data from a version known to contain an important bug when a corrected baseline simulation is available. I don't see why this cannot be excluded. - Previous literature has pointed to notable differences between the Pacific and Atlantic Ocean in model-measurement comparison, which I am concerned could be hidden here by merging across latitude bands. - It would be helpful with a clearer description of the relationship between SO2 and total particle number concentrations, specifically the role of uncertainties in other aerosol species for the latter.

Specific: Line 89: and by Katich et al. (2018) for AeroCom multi-model data and Lund et al. (2018) for two global models.

Line 100: previous studies have also investigated the role of spatial and temporal sampling specifically for aircraft campaigns, including ATom (Samset et al, Lund et al. 2018). Referring to these could be useful.

Line 120: Could there similarly be a problem with performing only one-at-a-time sensitivity tests in this regard, specifically if they are designed to perturbed parameters known to relate more to one of the three variables one wants to evaluate?

Line 140: is data from all four ATom campaigns lumped together in the comparison with modeled number concentrations? I.e., you're comparing quite different amounts of data points and there would be variations across ATom phases/seasons – does this influence the discussion/interpretation of SO2-particle number relationship?

Line 148: Are these present in addition to the ones above? Please clarify.

Line 163: I'm missing information about emissions used in the model, including natural sources (e.g., DMS, biomass burning). Would be useful to add.

Line 187-188: Is the model run with nudging for all three ATom years? Please clarify.

Line 220: it could be useful with some information about the particle size distribution in the baseline simulation as well.

Line 236: SO3. . .

Line 244: This confused me a bit. What is then shown in Figure 2 – w or wo error? Why do you want to show the erroneous model version at all? I think you need to show (in the appendix perhaps) the difference between the two and add a rationale as to why are you using the default version with the bug at all, and not just the new, correct baseline. If it's only for reference to CMIP6 data, I would suggest moving that to the Appendix all together to avoid it distracting from the current study. (The choice to show both becomes even more puzzling in Figure 4.)

Line 247: would be nice to add these region definitions to Figure 1.

Line 247: Previous studies of other variables/aerosols have shown that there can be significant differences in model-measurement agreement over the Pacific versus Atlantic Ocean in the ATom and HIPPO evaluations. I would encourage the authors to do a similar regional split of their current latitude bands to check if this is the case also for these variables. Might shed some new light and be useful for the community.

Line 260-264: If the bug-fix does not affect SO2, what is the reason for the small differences in the tropics?

Line 343: is it possible to from available literature to provide an assessment of the role of oceanic sources relative to terrestrial, in general? The lack of former here raises the question of what the result of the sensitivity test would be if you did include marine organics or whether that could reduce the bias on its own.

Line 493/494: Would it be possible to somehow distinguish positive and negative biases in Table 2 (and A2), as it's not easy to remember all the details of the results from the first comparison?

Line 532: The two condsink simulations are very pronounced on the bar chart and should be mentioned

Line 565: "can help to reduce" yes, but what is the realism? Given the opposite effect on condensation sink bias, I think this should be rephrased. Maybe say that Ntotal is very sensitive to this assumption or something.

Line 602: It would be great to already at the beginning of section 6 specify that a three-way comparison will be done in section 7. I was left wondering why the combined impacts on parameters were not discussed in the preceding sections.

Line 663: I.e., the NMBF as defined in section 4, or a new measure? Please clarify.

Line 687: I think this should also be mentioned at a much earlier stage

Line 692: it would be good to have a similar split between Northern and Southern hemisphere within the high lats and extratropics here, as for the baseline comparison. (see also my comment to line 244).

Line 713: could you give a couple of examples of such possible structural errors?

Line 746: consistent across latitude bands/regions as well?

---

## Referee Comment (RC2) · Anonymous Referee #2 · 24 Dec 2020

The authors present a comprehensive evaluation of the UKESM model against ATom particle number concentration, SO2 mixing ratio, and condensation sink data. The model includes state of the art aerosol microphysics packages and all the bells and whistles of CMIP6-era chemistry-climate modeling. The authors are quite thorough in their analysis, which makes for a satisfying, if rather long, paper. I especially like the various bar graphs and table which very clearly show which factors improve model bias. The paper is well written. Overall I think this is a solid paper that is worthy of publication subject to some minor revisions.

General comments:

1. I am not sure a bug fix is worthy of such attention as given in this paper. It would be equally good, in this reviewer's opinion, to simply start the results with analysis of the 'baseline' simulation, and briefly describe the bug fix. Perhaps the 'default' (buggy) model simulation results could be put in the supplement.

2. I was looking for more detailed information on the simulation setup and could not seem to find it. It is mentioned that the model is sampled for the ATom flight tracks and both the observations and model output are averaged up to (monthly?) means to beat down the noise. So how long was the model actually run for the baseline simulation and each of the many sensitivity simulations? Just the ATom period? 2016-2018? Which years, since some analysis uses ATom-4 only and others use ATom 1-4. This becomes more important for the figures that show vertical profiles of model vs measured: what exact time periods are we looking for here? This should be mentioned in the caption at least. I could not find this information (apologies if it is somewhere else).

3. How does the use of ATom-4 only for one evaluation metric versus all of ATom 1-4 for other metrics impact the results? The authors should comment a bit on that.

Specific comments: P2L36: Unclear what "Analyzing the data with altitude" means here, since the previous sentence does exactly that. What's different?

P2L40: The perturbations could be well motivated but we don't know really what they are in the abstract. Of course we learn later in the main text. I think a sentence somewhere saying briefly what they are would be helpful.

P2L50: Is aerosol size strictly an "optical" property?

P4L116: Is the citation of Adams and Seinfeld (2002) correct here? That is a model description paper of TOMAS. The sentence as written is about how this paper evaluates the model.

P16: Seems to be a microsoft "paste" icon somehow in this figure

P20L385: We need a mathematical equation for how condensation sink is calculated

[Figure]

P23L466: Consistent notation of numbers with exponents

P24L480: Odd to start discussing Figure 10 before discussing Figures 7-9.

---

## Author Comment (AC1) · 4 Feb 2021

**Response to reviewers**

We thank the reviewers for their critical feedback and insightful comments. We have responded to each of the comments in-line below and, where appropriate, indicated the relevant changes in the manuscript.

**Reviewer 1:**

**General:** - I'm puzzled by the use of model data from a version known to contain an important bug when a corrected baseline simulation is available. I don't see why this cannot be excluded. - Previous literature has pointed to notable differences between the Pacific and Atlantic Ocean in model-measurement comparison, which I am concerned could be hidden here by merging across latitude bands. - It would be helpful with a clearer description of the relationship between SO2 and total particle number concentrations, specifically the role of uncertainties in other aerosol species for the latter

We have addressed the issue of the default model and moved those plots to the Appendix figures A1,A2 and A3. New figures 3,4 and 5 deal exclusively with the baseline simulation and observations. New Appendix Figure A5 represents how the model performs in the Pacific and Atlantic ocean. To describe the role of $SO_2$ on particle number concentrations the quoted text is present in the section 1.

"Perturbing atmospheric processes can also have a direct effect on the $SO_2$ mixing ratio and affects $H_2SO_4$ concentration which controls new particle formation (NPF), and we know from past studies (Gordon et al., 2017) that new particle formation is the source of about half of the CCN in the atmosphere"

**Specific:**

Line 89: and by Katich et al. (2018) for AeroCom multi-model data and Lund et al. (2018) for two global models.

Done. We have included these suggested references in the manuscript.

Line 100: previous studies have also investigated the role of spatial and temporal sampling specifically for aircraft campaigns, including ATom (Samset et al, Lund et al. 2018). Referring to these could be useful.

Done. These references have been included in the updated manuscript

Line 120: Could there similarly be a problem with performing only one-at-a-time sensitivity tests in this regard, specifically if they are designed to perturbed parameters known to relate more to one of the three variables one wants to evaluate?

Yes, you are right, some parameters may affect one variable more than the other two. As an example, nucleation rate which affects the biases in $N_{Total}$ more than $SO_2$ or condensation sink. So reducing the nucleation rate in the upper troposphere helps reduce the positive $N_{Total}$ bias in the tropics but doesn't affect $SO_2$ and condensation sink. Apart from say nucleation rate and

primary aerosol emissions, most parameters affect all three variables to a reasonable extent. We acknowledge this in section 5 where we have described why we have chosen each of these parameters and speculate about how they could affect biases.

Line 140: is data from all four ATom campaigns lumped together in the comparison with modeled number concentrations? I.e., you're comparing quite different amounts of data points and there would be variations across ATom phases/seasons – does this influence the discussion/interpretation of SO2-particle number relationship?

Quoted text added to manuscript section 4 to address your point.

"The $SO_2$ instrument was only flown on the ATom4 campaign, in spring 2018, while the vertical profiles of $N_{Total}$ and Condensation sink are produced using all of the ATom campaigns, in all four seasons. However, we compare like with like, in that, for example, $SO_2$ observations in spring are compared only with $SO_2$ model data at three-hourly time resolution in spring. We perform our analysis using the available data, however our analysis could benefit from more $SO_2$ data. We also can see from the that the vertical profiles of $N_{Total}$ and condensation sink for just ATom 4 (Appendix figure A4) show similar biases as figure 3 and 5, which have data from all the ATom campaigns aggregated together. "

Line 148: Are these present in addition to the ones above? Please clarify

Yes, from ATom2 to 4 more CPC's were present in addition to those present in ATom1. We have added text in the manuscript to make this clearer.

Line 163: I'm missing information about emissions used in the model, including natural sources (e.g., DMS, biomass burning). Would be useful to add.

The quoted text has been added to the manuscript section 3 to reference the emission dataset used in the model.

"The anthropogenic, biomass burning , biogenic and DMS land emissions used by the model are taken from Hoesly et al 2018, Van Marle et al 2017, Sindelarova et al 2014 and Spiro et al., 1992 respectively"

Line 187-188: Is the model run with nudging for all three ATom years? Please clarify.

Yes, the model is nudged for the duration and time of each of atom campaigns so as to reproduce the meteorological conditions at which these measurements were taken. Text added to section 3.

"To compare the model against observations, we run the model in a nudged configuration over the period during which the ATom campaigns took place (2016-2018)."

Line 220: it could be useful with some information about the particle size distribution in the baseline simulation as well.

We hesitate to include this in the discussion simply to avoid making an already lengthy paper, lengthier. In this work we focus on $N_{Total}$ and condensation sink from which we can make inferences about the aerosol modes. Condensation sink is a suitable proxy for the accumulation and coarse mode. As an example when we perturb the Vehkamaki nucleation rate ($J_{veh}$) there is no significant change in the condensation sink but there is a significant change in $N_{Total}$. From this we can infer that smaller aerosol modes (nucleation/Aitken mode) contributes to the change in $N_{Total}$.

Line 236: SO3. . .

Done. Changed $SO_3$ to $SO_2$

Line 244: This confused me a bit. What is then shown in Figure 2 – w or wo error? Why do you want to show the erroneous model version at all? I think you need to show (in the appendix perhaps) the difference between the two and add a rationale as to why are you using the default version with the bug at all, and not just the new, correct baseline. If it's only for reference to CMIP6 data, I would suggest moving that to the Appendix all together to avoid it distracting from the current study. (The choice to show both becomes even more puzzling in Figure 4.)

New figures 3, 4 and 5 which show only the baseline version of the model are added to the manuscript. The plots of the default version and how it compares with the baseline version has been moved to the appendix figure A1, A2 and A3.

Line 247: would be nice to add these region definitions to Figure 1.

Done.

Line 247: Previous studies of other variables/aerosols have shown that there can be significant differences in model-measurement agreement over the Pacific versus Atlantic Ocean in the ATom and HIPPO evaluations. I would encourage the authors to do a similar regional split of their current latitude bands to check if this is the case also for these variables. Might shed some new light and be useful for the community.

A new figure has been added to the Appendix (Figure A5) to illustrate the differences between the Pacific and Atlantic. There are differences in magnitude between the Pacific and Atlantic and the model captures the same trends when compared to observations and shows similar biases in both regions. The following text in quotes has been added to section 4 of the manuscript

"To explore any longitudinal differences, we also plotted the observations and model data in the Pacific and Atlantic Ocean to briefly explore whether the model shows differing trends in these regions (Appendix Figure A5). From the figure we can see that the model shows biases of similar magnitude in the Pacific and Atlantic when compared to observations. The model shows biases of up to 10, 5 and 2 for the $N_{Total}$, $SO_2$ and condensation sink respectively in the Pacific and Atlantic"

Line 260-264: If the bug-fix does not affect SO2, what is the reason for the small differences in the tropics?

Done. Quoted text added to section 4

"We speculate that the small differences in biases we see between the baseline and default version (Figure A2) are due to cloud adjustments, which can affect the $SO_2$ concentration and condensation sink. Adjustments arise because changes in $N_{Total}$ can affect cloud drop concentration and liquid water path, and can therefore change the $SO_2$ lost in aqueous chemical processing in clouds"

Line 493/494: Would it be possible to somehow distinguish positive and negative biases in Table 2 (and A2), as it's not easy to remember all the details of the results from the first comparison?

Done.

Line 532: The two condsink simulations are very pronounced on the bar chart and should be mentioned

Done. I added the quoted text to section 6.1

"The simulations where we increase the condensation sink by a factor of 5 and 10 show larger biases (NMAEF = 2.46 and 5.5 respectively). These perturbations are somewhat unrealistic, because the baseline version already agrees well (within a factor of 2) with observations, but they are useful as tests of the sensitivity of new particle formation in the model to the condensation sink."

Line 565: "can help to reduce" yes, but what is the realism? Given the opposite effect on condensation sink bias, I think this should be rephrased. Maybe say that Ntotal is very sensitive to this assumption or something.

Done. I added the quoted text to section 6.2

"However, as noted earlier, directly scaling the condensation sink by factors of 5 and 10 in this way is unrealistic, as the model's condensation sink is within a factor of 2 of observations (Figure 6)"

Line 602: It would be great to already at the beginning of section 6 specify that a threeway comparison will be done in section 7. I was left wondering why the combined impacts on parameters were not discussed in the preceding sections.

Done. I added the text in quotes to the beginning of section 5 instead of 6.

"a three way comparison of $N_{Total}$, $SO_2$ and condensation sink biases is explored in Section 7"

Line 663: I.e., the NMBF as defined in section 4, or a new measure? Please clarify

NMAEF was used. I have clarified this in the manuscript as well.

Line 687: I think this should also be mentioned at a much earlier stage

Yes the effects of Cloud pH on upper tropospheric model performance has been elaborated in section 6.2

Line 692: it would be good to have a similar split between Northern and Southern hemisphere within the high lats and extratropics here, as for the baseline comparison. (see also my comment to line 244).

I have included Figure A9 to the appendix to address how the combined simulation performs in both hemispheres. The text in quotes has been added to the manuscript section 7.2

"The interhemispheric differences in the vertical profile of the combined simulation and baseline simulation are shown in the Appendix (Figure A9). Overall, the combined simulation performs better than the baseline simulation in both hemispheres, with a couple of notable exceptions. The combined simulation underpredicts observations of $N_{Total}$ in the southern high latitude upper troposphere and of $SO_2$ concentration in the northern high latitude upper troposphere by up to a factor of 2 more than the baseline simulation. We speculate that a marine nucleation mechanism or regional changes in cloud pH that are not simulated in the model currently could be the reason for interhemispheric biases."

Line 713: could you give a couple of examples of such possible structural errors?

Quoted text added to section 7

"Clearly structural errors in the model remain, possibly uncertainties associated with the convection parametrisation (Prein et al., 2015) or other atmospheric processes"

Line 746: consistent across latitude bands/regions as well?

Yes, this has been explained in section 7.2. I have also added an additional figure (Figure A8) looking at how the combined simulation performs in each hemisphere.

**Reviewer 2:**

**General Comments**

1. I am not sure a bug fix is worthy of such attention as given in this paper. It would be equally good, in this reviewer's opinion, to simply start the results with analysis of the 'baseline' simulation, and briefly describe the bug fix. Perhaps the 'default' (buggy) model simulation results could be put in the supplement.

Done. I have changed figures 3,4 and 5 which show only the baseline simulation and observations. I have moved the plots with the default version into the appendix (Appendix Figure A1, A2 and A3)

2. I was looking for more detailed information on the simulation setup and could not seem to find it. It is mentioned that the model is sampled for the ATom flight tracks and both the

observations and model output are averaged up to (monthly?) means to beat down the noise. So how long was the model actually run for the baseline simulation and each of the many sensitivity simulations? Just the ATom period? 2016-2018? Which years, since some analysis uses ATom-4 only and others use ATom 1-4. This becomes more important for the figures that show vertical profiles of model vs measured: what exact time periods are we looking for here? This should be mentioned in the caption at least. I could not find this information (apologies if it is somewhere else).

The model is set up to output data at high temporal resolution (3 hourly) to reduce sampling errors, and the three-hourly model output is interpolated onto each 5-minute-averaged point along the flight track for an accurate comparison. The vertical profiles are then aggregates of these interpolated values. No monthly averages are calculated. In Figures 3,4 and 5, I have mentioned the campaigns that were used to produce those plots. Quoted text added to manuscript section 4.

"The $SO_2$ instrument was only flown on the ATom4 campaign, in spring 2018, while the vertical profiles of $N_{Total}$ and Condensation sink are produced using all of the ATom campaigns, in all four seasons. However, we compare like with like, in that, for example, $SO_2$ observations in spring are compared only with $SO_2$ model data at three-hourly time resolution in spring. We perform our analysis using the available data, however our analysis could benefit from more $SO_2$ data. We also can see from the that the vertical profiles of $N_{Total}$ and condensation sink for just ATom 4 (Appendix figure A4) show similar biases as figure 3 and 5 which have data from all the ATom campaigns aggregated together."

3. How does the use of ATom-4 only for one evaluation metric versus all of ATom 1-4 for other metrics impact the results? The authors should comment a bit on that.

Done. Addressed in previous point.

Specific Comments:

P2L36: Unclear what "Analyzing the data with altitude" means here, since the previous sentence does exactly that. What's different?

Done. Correction made to manuscript. You are right, I did do that in the previous 2 sentences in the manuscript.

P2L40: The perturbations could be well motivated but we don't know really what they are in the abstract. Of course we learn later in the main text. I think a sentence somewhere saying briefly what they are would be helpful.

Done. Quoted text added to Abstract

"The perturbations take the form of global scaling factors or improvements to the representation of atmospheric processes in the model, for example by adding a new boundary layer nucleation scheme"

P2L50: Is aerosol size strictly an "optical" property?

You are right its more of a morphological property rather than an optical property. I have corrected "aerosol optical properties" to just "aerosol properties".

P4L116: Is the citation of Adams and Seinfeld (2002) correct here? That is a model description paper of TOMAS. The sentence as written is about how this paper evaluates the model.

Done. Removed the citation

P16: Seems to be a microsoft "paste" icon somehow in this figure

Yes, I have changed the figures.

P20L385: We need a mathematical equation for how condensation sink is calculated

Done. Added the following text

"The condensation sink refers to the rate at which these condensable gases condense onto aerosol particles in the atmosphere. It is equal to $2\pi D \sum_j \beta_j d_j N_j$, where D is diffusion coefficient, $\beta_j$ is the transition regime correction factor (Fuchs and Sutugin, 1971), $d_j$ is the particle diameter and Nj is the particle number concentration for the jth aerosol mode."

P23L466: Consistent notation of numbers with exponents

Done

P24L480: Odd to start discussing Figure 10 before discussing Figures 7-9.

Done